# Site-specific spectroscopic measurement of spin and charge in $(LuFeO_3)_m/(LuFe_2O_4)_1$ multiferroic superlattices

Shiyu Fan [1], Hena Das[2,3,4], Alejandro Rébola [5], Kevin A. Smith [6], Julia Mundy [2,7], Charles Brooks[8], Megan E. Holtz [2,8], David A. Muller [2,9], Craig J. Fennie [2], Ramamoorthy Ramesh [10,11,12], Darrell G. Schlom [8,9], Stephen McGill[13] & Janice L. Musfeldt [1,6✉]

Interface materials offer a means to achieve electrical control of ferrimagnetism at room temperature as was recently demonstrated in $(LuFeO_3)_m/(LuFe_2O_4)_1$ superlattices. A challenge to understanding the inner workings of these complex magnetoelectric multiferroics is the multitude of distinct Fe centres and their associated environments. This is because macroscopic techniques characterize average responses rather than the role of individual iron centres. Here, we combine optical absorption, magnetic circular dichroism and first-principles calculations to uncover the origin of high-temperature magnetism in these superlattices and the charge-ordering pattern in the $m = 3$ member. In a significant conceptual advance, interface spectra establish how Lu-layer distortion selectively enhances the $Fe^{2+} \rightarrow Fe^{3+}$ charge-transfer contribution in the spin-up channel, strengthens the exchange interactions and increases the Curie temperature. Comparison of predicted and measured spectra also identifies a non-polar charge ordering arrangement in the $LuFe_2O_4$ layer. This site-specific spectroscopic approach opens the door to understanding engineered materials with multiple metal centres and strong entanglement.

[1] Department of Physics and Astronomy, University of Tennessee, Knoxville, TN 37996, USA. [2] School of Applied and Engineering Physics, Cornell University, Ithaca, NY 14853, USA. [3] Laboratory for Materials and Structures, Tokyo Institute of Technology, Midori-ku, 4259 Nagatsuta, Yokohama, Kanagawa 226-8503, Japan. [4] Tokyo Tech World Research Hub Initiative, Institute of Innovative Research, Tokyo Institute of Technology, 4259 Nagatsuta, Midori-ku, Yokohama, Kanagawa 226-8503, Japan. [5] Instituto de Física Rosario-CONICET, Boulevard 27 de Febrero 210 bis, 2000 Rosario, Argentina. [6] Department of Chemistry, University of Tennessee, Knoxville, TN 37996, USA. [7] Department of Physics, Harvard University, Cambridge, MA 02138, USA. [8] Department of Materials Science and Engineering, Cornell University, Ithaca, NY 14853, USA. [9] Kavli Institute at Cornell for Nanoscale Science, Ithaca, NY 14853, USA. [10] Department of Materials Science and Engineering, University of California, Berkeley, CA 94720, USA. [11] Department of Physics, University of California, Berkeley, CA 94720, USA. [12] Materials Sciences Division, Lawrence Berkeley National Laboratory, Berkeley, CA 94720, USA. [13] National High Magnetic Field Laboratory, Tallahassee, FL 32310, USA. ✉email: musfeldt@utk.edu

The dream of a ferroelectric ferromagnet that is fully coupled at room temperature is the grand challenge of multiferroics and magnetoelectrics. Heteroepitaxy enlarges the design space to achieve this difficult but important goal, and examples abound of superlattices and interfaces at which exotic properties emerge[1–19]. Superlattices of the form $(LuFeO_3)_m/(LuFe_2O_4)_n$ that sport ferroelectric ferrimagnetism are prominent examples[20]. The layer indices run from 0 to 9 and for simplicity are denoted $(m, n)$. One end member $h$-$LuFeO_3$ is a polar, improper ferroelectric below 1020 K, and it orders anti-ferromagnetically at 147 K in a pattern in which symmetry allows a slight canting of the spins—giving rise to weak ferromagnetism[21–24]. The other end member $LuFe_2O_4$ is an anti-ferroelectric with a complex phase diagram, exemplified by a series of charge-ordering transitions above room temperature, a 240 K ferrimagnetic ordering temperature, and a structural transition near 170 K[25–30]. The crystal structures of the end members are shown in Supplementary Fig. 1. In certain members of the $(LuFeO_3)_m/(LuFe_2O_4)_n$ series, ferroelectric ferrimagnetism emerges with ordering temperatures up to 281 K[20]. Such a superlattice has a higher magnetic ordering temperature than either of its end members due to interface effects[20]. The microscopic nature of these interface effects and their connection to the more robust magnetism is highly under explored. At the same time, the charge-ordering pattern in the $LuFe_2O_4$ layer is complex, with both polar and nonpolar Fe double layers predicted to be essentially isoenergetic[20,25,26]. The symmetric vs. asymmetric displacement of the upper and lower Lu layers adjacent to $LuFe_2O_4$ is one characteristic that differentiates the polar vs. nonpolar Fe double layer charge-ordering candidates. Resolving these issues is crucial for determining the ground state. Here, we use a spectroscopic approach that builds upon magnetic property, polarization, and neutron scattering measurements[20,21,23] to reveal the site-specific electronic structure of these engineered materials, unraveling both the microscopic origin of high-temperature magnetism in the $(LuFeO_3)_m/(LuFe_2O_4)_n$ family of materials and the charge-ordering pattern in the Fe bilayer of the (3, 1) superlattice. In addition to introducing a remarkably powerful and versatile technique for extracting spin and charge character at the interface in a homologoes series of multiferroic heterostructures, our work opens the door to similar approaches in other engineered materials as well as opportunities for the development of structure–property relationships and interface descriptors, potentially advancing a number of allied fields including spintronics and photonics.

## Results and discussion

### Uncovering the electronic excitations of different Fe centers in the (3, 1) superlattice.

We begin with the (3, 1) superlattice because it is the most theoretically tractable. Figure 1a, b displays the crystal structure highlighting the $(LuFeO_3)_3/(LuFe_2O_4)_1$ layer pattern along with a scanning transmission electron microscope (STEM) image of the film. Inversion symmetry in the $LuFe_2O_4$ layer is broken due to the rumpling imposed by the adjacent $LuFeO_3$ layers[20]. This is because the pattern of Lu-layer distortions around the Fe double layer is asymmetric with both down/up/up and down/up/down displacements along $c$. Here, $d$ represents the size of the Lu-layer displacement. One way to separate the role of the different metal sites—at least in principle—is by projecting out the contribution of various layers and their Fe centers. Figure 1c displays the spin-projected density of states for the Fe double layer in $LuFe_2O_4$, the adjacent monolayer and the central monolayer in $LuFeO_3$. The six types of excitations (summarized in Table 1) provide for a site-specific analysis of magnetism in $(LuFeO_3)_3/(LuFe_2O_4)_1$. This is an unusual amount

of complexity for dichroic analysis of an iron-containing material. Fortunately, of these six excitations, only three are important due to the relative size of the matrix elements. For instance, the two charge-transfer excitations are quite strong in the linear absorption and magnetic circular dichroism because they involve Fe sites with different charges. The on-site $Fe^{2+}$ $d$-to-$d$ excitation is important in the dichroic response due to the large $Fe^{2+}$ density of states in the spin-down channel (Fig. 1c). This feature is also evident in the optical absorption of the $LuFe_2O_4$ end member[27]. That these structures occur in different energy regions allows us to separate closely related Fe-containing excitations in both the optical absorption and magnetic circular dichroism.

Figure 2a, b summarizes the spectroscopic response of $(LuFeO_3)_3/(LuFe_2O_4)_1$. The challenge that arises immediately — even upon cursory inspection of the dichroic spectra—is how to distinguish the different Fe contributions. One path forward is to employ the linear absorption spectrum, $\alpha(E)$, along with assignments from electronic structure calculations[20–22,27,31,32] to determine characteristic excitation energies of each type of iron center. Figure 2a displays the optical absorption spectrum of the (3, 1) superlattice. Based upon our spin-projected density of states calculations (Fig. 1c and Table 1), the Fe-related excitations take place over a broad energy range. We can address the various site-specific Fe-related excitations by dividing the spectra into different energy regions and performing subtractions where necessary. As a reminder, the most important are (i) the $Fe^{2+} \rightarrow Fe^{3+}$ charge-transfer excitations in the spin-up and spin-down channels of the $LuFe_2O_4$ double layer and (ii) the $Fe^{2+} d \rightarrow d$ on-site excitation in the spin-down channel of the $LuFe_2O_4$ double layer. These features are indicated in Fig. 2a.

Figure 2b displays the magnetic circular dichroism spectrum of the (3, 1) superlattice at full field. Access to the 25 T split helix magnet at the National High Magnetic Field Laboratory[33] was crucial to this work, providing both direct optical access and a field high enough to saturate the magnetic state of interest. Two zero-field spectra are also included. They are not the same because the measurement pathway is hysteretic (25 T $\rightarrow 0^-$ T $\rightarrow -25$ T $\rightarrow 0^+$ T $\rightarrow 25$ T), and the ferrimagnetic film is not fully demagnetized when the field is removed. At full field, the dichroic spectra reveal a broad, asymmetric structure centered at 1.5 eV and a smaller lobe near 2.2 eV. Based upon Fig. 2a, we assign the 1.5 eV feature to the $Fe^{2+} \rightarrow Fe^{3+}$ charge-transfer excitation of the $LuFe_2O_4$ layer in the spin-down channel. The small 2.2 eV feature has a more complex origin because excitations are heavily mixed in this energy range. Our analysis shows that this structure emanates from a combination of charge-transfer excitations in both the spin-down and spin-up channels. The sign change at 2.1 eV is a reminder of how the spin-up channel density of states comes to dominate the response. There is another inflection point near 2.4 eV, above which $\Delta\alpha$ changes sign due to the way in which the spin-down channel $Fe^{2+} d \rightarrow d$ excitation dominates the dichroic response. The on-site $d$-to-$d$ excitations of $Fe^{3+}$ have much lower intensity (Table 1). Importantly, features in the dichroic spectra $\Delta\alpha(E)_{MCD}$ are directly proportional to net magnetization, and since we can analyze this effect at different energies, the response can be correlated with specific iron centers[34–36].

### Revealing the role of each individual Fe center.

In order to uncover the role of each type of Fe center, we take constant energy cuts of the dichroic spectra based upon the excitation of interest. For instance, constant energy cuts of the spectra at 1.33 eV reveal the behavior of the $Fe^{2+} \rightarrow Fe^{3+}$ charge-transfer excitation in the spin-down channel. Remarkably, a plot of $\Delta\alpha_{MCD}$ at

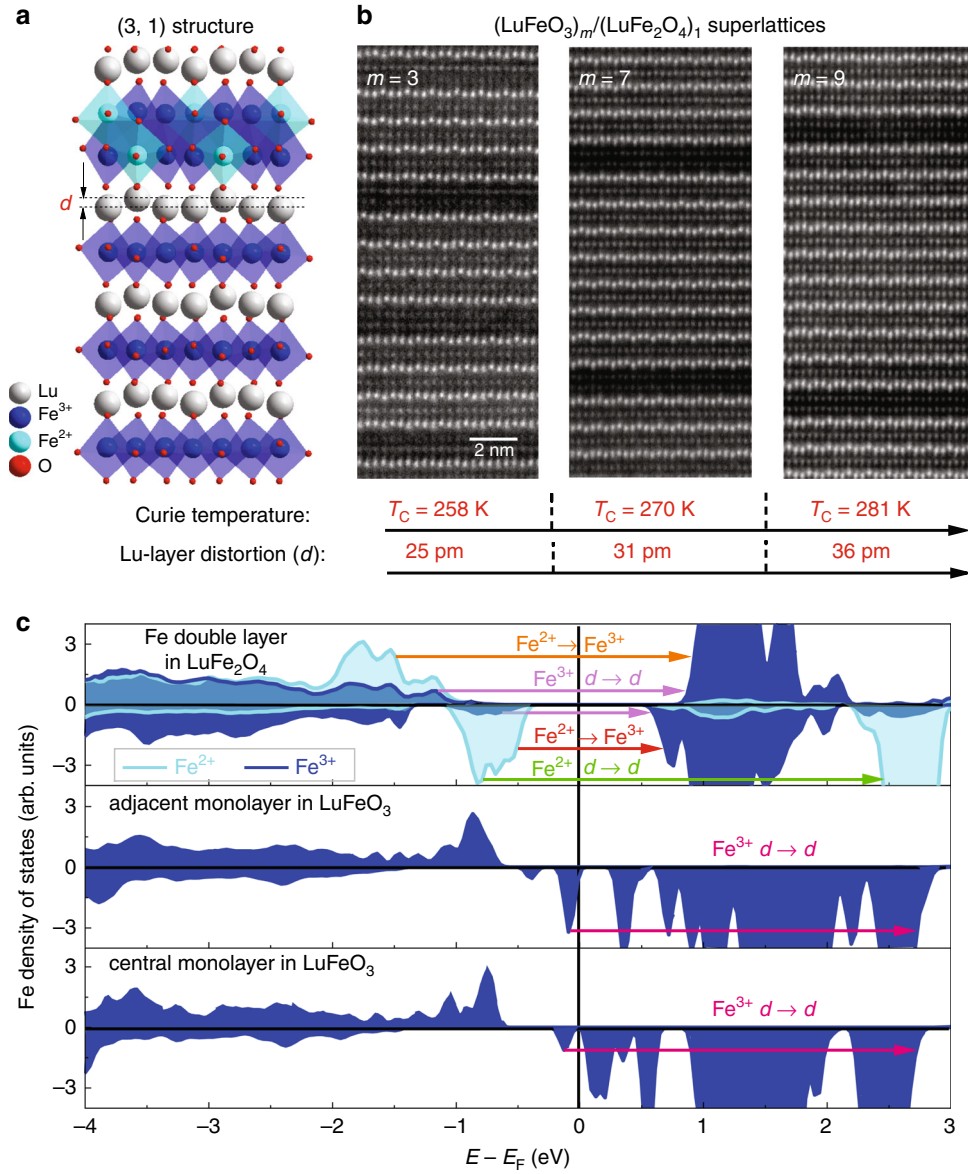

**Fig. 1 Crystal structure, growth pattern, and spin-projected density of states. a** Crystal structure of the (3, 1) film showing how a $LuFe_2O_4$ slab (which has an iron oxide double layer with both $Fe^{2+}$ and $Fe^{3+}$ between two Lu layers) is sandwiched between three layers of $LuFeO_3$[20]. $d$ represents the Lu-layer displacement. **b** HAADF-STEM images of the $m = 3$, 7, and 9 superlattices viewed along the [$1\bar{1}0$] zone axis. Atomic number contrast shows the bright, heavy lutetium atomic rows layered with the less bright iron atomic rows. The scale bar is the same for all images. Trends in the ferrimagnetic $T_C$ and Lu-layer distortion (which increases with the number of $LuFeO_3$ layers) are also shown. **c** Spin-projected density of states of the Fe double layer in $LuFe_2O_4$, an adjacent monolayer of $LuFeO_3$ and the central $LuFeO_3$ monolayer. These calculations were performed using the self-doped charge-ordering model as discussed in the text (Fig. 4b). The Fe double layer is nonpolar, and the Lu-layer displacement is asymmetric with both down/up/up and down/up/down distortion patterns around the Fe bilayer. The $Fe^{3+}$ and $Fe^{2+}$ states are indicated with dark and light blue, respectively. The arrows denote different types of excitations.

**Table 1 Summary of different types of Fe excitations in the (3, 1) superlattice.**

| Type of excitation | Excitation channel | Layer in the superlattice | Energy range (eV) | Intensity ($\Delta\alpha(E)$, $\alpha(E)$) |
|---|---|---|---|---|
| $Fe^{2+} \rightarrow Fe^{3+}$ charge transfer | Spin-down | $LuFe_2O_4$ | 1–2.4 | strong, strong |
| $Fe^{2+} \rightarrow Fe^{3+}$ charge transfer | Spin-up | $LuFe_2O_4$ | 2–2.8 | strong, strong |
| $Fe^{2+}d \rightarrow d$ on-site | Spin-down | $LuFe_2O_4$ | 2.5–2.8 | medium, medium |
| $Fe^{3+}d \rightarrow d$ on-site | Spin-up | $LuFe_2O_4$ | 2–2.8 | weak, weak |
| $Fe^{3+}d \rightarrow d$ on-site | Spin-down | $LuFe_2O_4$ | 2–2.8 | weak, weak |
| $Fe^{3+}d \rightarrow d$ on-site | Spin-down | $LuFeO_3$ | 1–2.8 | weak, weak |

The first three rows indicate the most important excitations as discussed in Supplementary Information. The energy range for our measurements is from 0.8 - 2.8 eV.

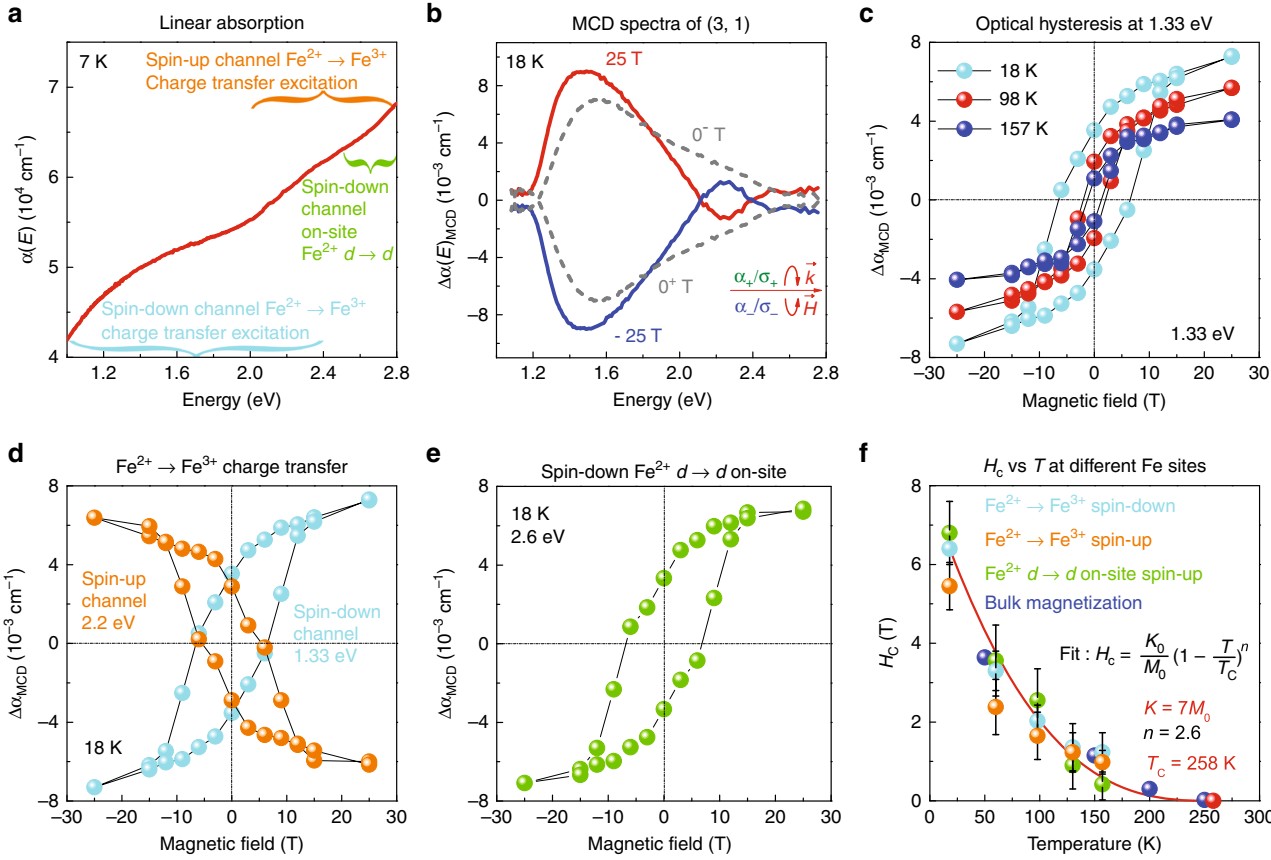

**Fig. 2 Linear absorption spectrum, magnetic circular dichroism, and the magnetic behavior of different Fe centers in $(LuFeO_3)_3/(LuFe_2O_4)_1$. a** Linear absorption spectrum of the (3, 1) superlattice. The three most important Fe-related excitations are indicated. **b** Magnetic circular dichroism spectra of the (3, 1) superlattice at ±25 and ±0 T after substrate correction. **c** Fixed energy cut of the magnetic circular dichroism spectrum at 1.33 eV as a function of magnetic field at various temperatures. **d** Optical hysteresis loop obtained from the analysis of the $Fe^{2+} \rightarrow Fe^{3+}$ charge-transfer excitation in the spin-up channel compared to that in the spin-down channel. **e** Hysteretic behavior of the $Fe^{2+}$ on-site $d$-to-$d$ excitation in the spin-down channel. **f** Coercive fields extracted from the optical hysteresis loops for each type of excitation as a function of temperature. The model fit is described in the text. Bulk magnetization is included for comparison[21].

1.33 eV vs. magnetic field unveils an optical hysteresis loop (Fig. 2c). Because the charge-transfer excitation is located in the $LuFe_2O_4$ layer, we can explicitly connect the behavior of the Fe bilayer to the magnetic response. We also measure the dichroic spectra of the (3, 1) superlattice at different temperatures. Analysis again reveals optical hysteresis loops that close with increasing temperature (Fig. 2c).

We also consider how other Fe centers support high-temperature magnetism in the (3, 1) superlattice by taking cuts of $\Delta\alpha_{MCD}$ at several different energies (Fig. 2d, e). While the optical hysteresis loop at 1.33 eV has a traditional shape, the loop becomes irregular at higher energies due to mixing (Supplementary Fig. 4b, c). The challenge is to extract the response of each individual Fe center from the mixed state. We perform constant energy cuts at 1.8, 2.2, and 2.6 eV to address this issue. Details are discussed in the Supplementary Information (Supplementary Fig. 4). By subtracting the dichroic response at 1.8 eV from that at 2.2 eV, we can obtain the pure signal of the charge-transfer excitation in the spin-up channel (Fig. 2d). The direction of the hysteresis loop thus obtained is reversed because the spin-state changes from down to up. A similar analysis is applied to the 2.6 eV energy cut of the spectral data. Here, spin-up charge transfer is strongly mixed with spin-down $Fe^{2+} d$-to-$d$ on-site excitations. Subtraction yields the signature of the $Fe^{2+}$ site (Fig. 2e). Note that the shape of the hysteresis loop returns to "normal" because the spin state flips again.

In order to link the microscopic response of the spin in the Fe double layer with the bulk magnetic properties[20,21], we extract the coercive fields from the optical hysteresis loops (Fig. 2c–e) and plot these spectroscopically determined coercive fields ($H_c$) with those obtained from bulk magnetization, as a function of temperature (Fig. 2f). The trend in $H_c$ is similar for all Fe centers, and the extracted coercive fields are in excellent agreement with bulk magnetization[20,21]. This demonstrates that a significant portion of the magnetism in the (3, 1) superlattice originates from the $LuFe_2O_4$ layer. In other words, the global coercive field is approximately equal to the local coercive field in the $LuFe_2O_4$ layer. We fit the temperature dependence of the coercive field in the (3, 1) superlattice with the Néel relaxation and Bean–Livingston models[37,38], which relate $H_c$ to the single-ion anisotropy ($K$), ferrimagnetic $T_C$, and the power index $n$ (Fig. 2f). Overall, this model is in reasonable agreement with our data, although $n = 2.6$ may indicate a slightly nonclassical response.

**Structure–property relations in the $(LuFeO_3)_m/(LuFe_2O_4)_n$ superlattices.** In order to unravel the mechanism of high-temperature magnetism and the consequences of Lu-layer distortion on the electronic structure of the interface, we measured the dichroic response of the (7, 1) and (9, 1) superlattices and compared the results to those of the (3, 1) material. As a

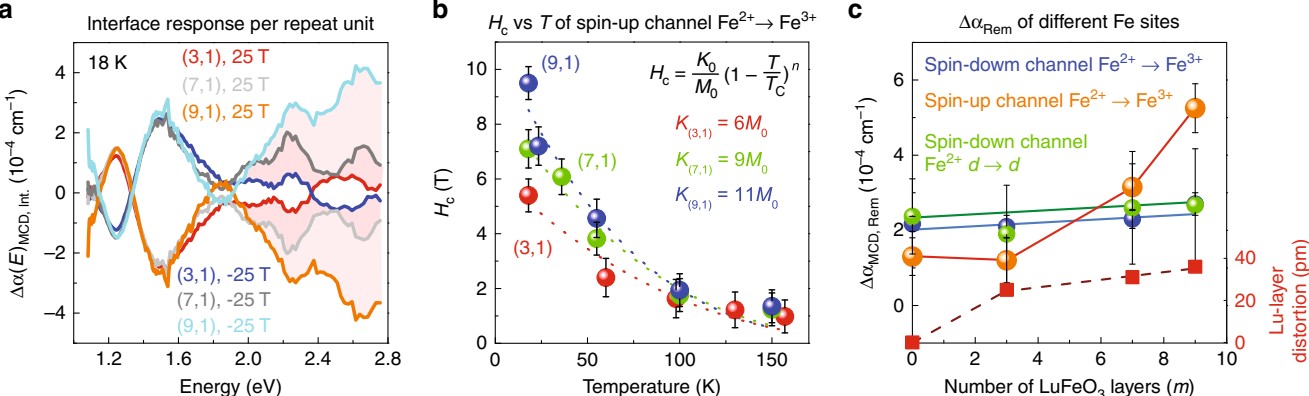

**Fig. 3 Magnetic circular dichroism of the interfaces, coercive fields, and remnant magnetization. a** Magnetic circular dichroism spectra of the interfaces on a "per repeat unit" basis. **b** Coercive fields obtained from analysis of the spin-up channel $Fe^{2+} \to Fe^{3+}$ charge-transfer excitation in the (3, 1), (7, 1), and (9, 1) superlattices vs. temperature. **c** Remanent magnetic circular dichroism for different types of Fe-related excitations and Lu-layer distortion vs. the number of $LuFeO_3$ layers ($m$). The Lu-layer distortion is taken from ref. [20], where total displacement $d = 1.5 \times Q$ the distortion amplitude [44]. See Table 1 and associated text for exact energies.

reminder, higher-order superlattices contain more $LuFeO_3$ layers, which increases the amplitude of the asymmetric Lu-layer distortion and raises the Curie temperature $T_C$ (Fig. 1). In order to make the most effective comparison, we need to isolate the spectroscopic response of the interface. We begin by normalizing the magnetic circular dichroism spectra on a "per repeat unit" basis by taking the raw dichroic signal and dividing by the number of repeat layers (Eq. (3)). Next, we use the normalized spectra of two end members to construct a "composite spectrum" for each superlattice. The composite response is simply the dichroic signal generated from combining the per repeat unit end member spectra based upon the composition, as given by $m$ and $n$ (Eq. (4)). We extract the interface spectrum of each film by subtracting the composite response from the measured spectrum on a "per repeat unit" basis. A detailed discussion of this procedure is available in the "Methods" section and Supplementary Information. We immediately see that the interface spectra, $\Delta\alpha_{MCD, Int.}$, is significant—at least at certain energies. This indicates that additional magnetism arises from the $LuFe_2O_4$–$LuFeO_3$ layer interaction.

Figure 3a summarizes the interface response of our set of superlattices. Remarkably, the interface spectra are nearly identical below 2 eV, demonstrating that magnetism emanating from the $Fe^{2+} \to Fe^{3+}$ charge-transfer excitation in the spin-down channel is only minimally dependent upon the size of the Lu-layer distortion (or the number of $LuFeO_3$ layers in the superlattice). The situation is different above 2 eV where, despite mixing with the spin-down channel transitions, the charge-transfer excitation in the spin-up channel dominates the dichroic response (Fig. 1c). This reveals that increased Lu-layer distortion selectively enhances the magnetic moment emanating from the spin-up channel $Fe^{2+} \to Fe^{3+}$ charge-transfer excitation, which amplifies the $LuFe_2O_4$ layer magnetization, and therefore the dichroic signal. This analysis naturally raises the question of exactly how Lu-layer distortion impacts individual Fe sites in the $LuFe_2O_4$ bilayer. We unveil the interface behavior of each Fe center by taking constant energy cuts of $\Delta\alpha_{MCD, Int.}$ and plotting these values as a function of magnetic field. In addition to coercivity and related trends in the single-ion anisotropy (Fig. 3b), the optical hysteresis loops that we extract from the magnetic circular dichroism spectra of the interface yield a remnant value of the dichroism ($\Delta\alpha_{MCD, rem.}$) that is proportional to remnant magnetization (Supplementary Fig. 4d–f). We can therefore reveal how superlattice periodicity affects local Fe site magnetization. Figure 3c displays $\Delta\alpha_{MCD, rem.}$ for the different Fe-related

excitations as a function of the number of $LuFeO_3$ layers. Because superlattice periodicity and the Lu-layer distortion are correlated, there is a relationship between $\Delta\alpha_{MCD, rem.}$ and the Lu-layer distortion as well. Above $m = 3$, the remnant signal from the charge-transfer excitation in the spin-up channel increases sharply—consistent with the theoretically predicted saturation moment in the $LuFe_2O_4$ layer [20]. By contrast, spin-down charge transfer and the $Fe^{2+} d \to d$ excitation are relatively insensitive to the number of $LuFeO_3$ layers (and the Lu-layer distortion). This behavior demonstrates that increasing magnetic moment in the $LuFe_2O_4$ layer emanates from rising $Fe^{2+}$ and $Fe^{3+}$ density of states in the spin-up channel of the higher-order superlattices. This conclusion arises from the corresponding changes in the dichroic spectra. At the same time, the trend provides a microscopic explanation for how high-temperature magnetism in these superlattices derives from Lu-layer distortion, as well as the more growth-oriented parameter of superlattice periodicity. We can understand in part why the enhanced magnetic moment emanates from the spin-up channel excitations by considering the charge-ordered state in greater detail.

**Determining the charge-ordering pattern in $(LuFeO_3)_3/(LuFe_2O_4)_1$.** Because charge ordering is one of the highest energy scales in the system [20,25–27], magnetism in the $(LuFeO_3)_m/(LuFe_2O_4)_1$ superlattices depends intimately upon the charge-ordering pattern in the Fe double layer. In order to reveal the relative importance of these states and distinguish between them, we calculated magnetic circular dichroism of several different candidate charge-ordering patterns, using first-principles methods and compared the results to the experimental dichroic spectra. The (3, 1) material has a large supercell containing 132 atoms, so we began by testing our predictions against the end members. Importantly, we tested two different states for the $LuFe_2O_4$ parent compound: CO-I and CO-II. Here, CO-I is an antiferroelectric state in which Lu trimer distortion is forbidden by symmetry (Supplementary Fig. 6a) [25,39]. CO-II, on the other-hand, allows Lu trimer distortion and has alternate $Fe^{2+}$- and $Fe^{3+}$-rich layers, the stacking of which breaks inversion and introduces ferroelectricity (Supplementary Fig. 6b) [40]. As discussed in the Supplementary Information, the computed spectra of both $LuFe_2O_4$ and $LuFeO_3$ are in good agreement with our measurements (Supplementary Fig. 7a, b), and a CO-I pattern is identified in the $LuFe_2O_4$ end member [25,39]. We therefore extended this approach to the (3, 1) superlattice.

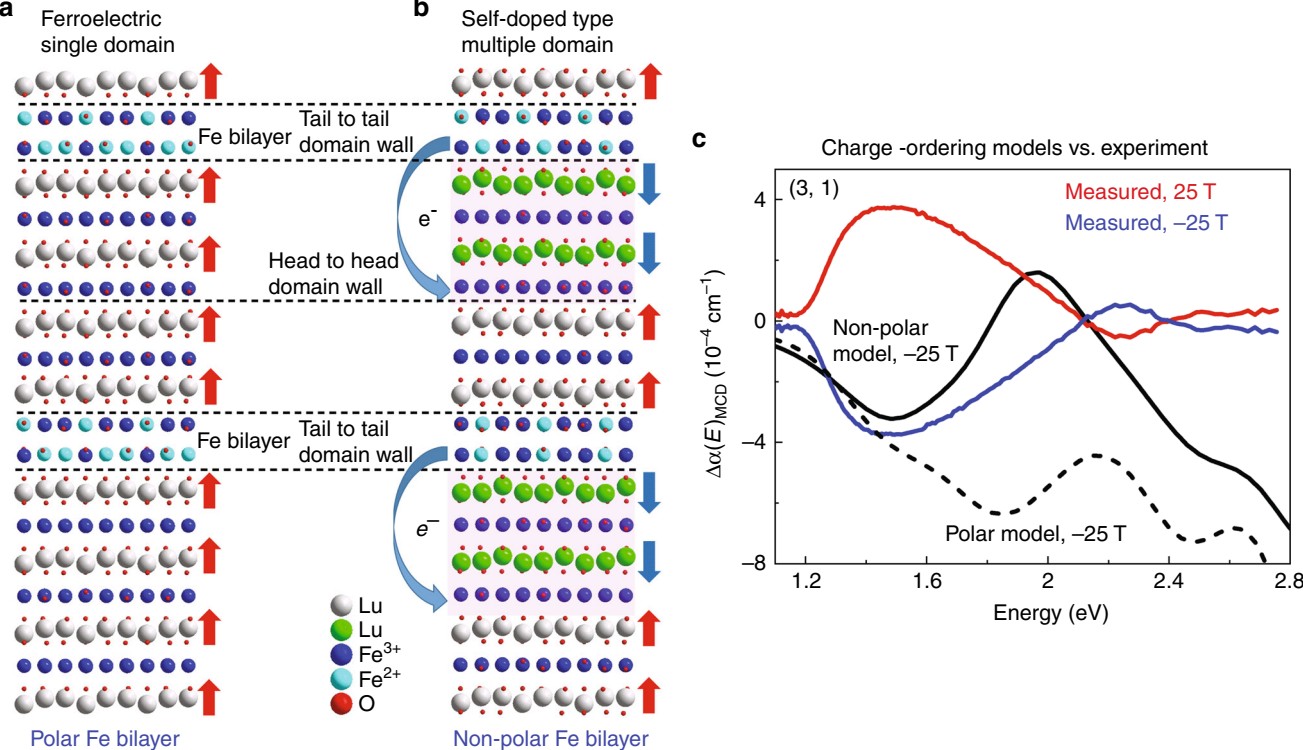

**Fig. 4 Candidate charge-ordering patterns for (LuFeO₃)₃/(LuFe₂O₄)₁ and comparison of calculated vs. measured dichroic spectra. a** Ferroelectric single-domain type (CO-FE) charge-ordered state with a polar Fe bilayer. The Lu-layer displacement is symmetric, and red arrows indicate the polarization direction. **b** Self-doped-type (CO-DOPED) multi-domain charge-ordered state with a nonpolar Fe bilayer, unveiling spontaneous electron transfer from the Fe bilayers to the LuFeO₃ layers. The polarization changes direction across the domain boundaries (dashed lines) as shown by the red and blue net dipole arrows, and the Lu-layer displacement is asymmetric with both down/up/up and down/up/down Lu distortion patterns surrounding the Fe double layer. **c** Experimental dichroic spectra $\Delta\alpha(E)_{MCD}$ along with calculated $\Delta\alpha(E)_{MCD}$ of the CO-FE (polar Fe bilayer) and CO-DOPED (nonpolar Fe bilayer) charge-ordering states in the (3, 1) superlattice.

The charge-ordered state in (LuFeO₃)₃/(LuFe₂O₄)₁ is more complicated than that in the LuFe₂O₄ end member due to Lu-layer distortion at the interface, which is induced by the LuFeO₃ layer. As a result, the CO-II-type state is more stable than CO-I for $m \geq 3$ superlattices. Based on a CO-II arrangement in the LuFe₂O₄ bilayer, theory predicts two possible charge-ordering patterns for the (3, 1) superlattice[20]. These candidates, termed CO-FE and CO-DOPED for reasons that will become clear below, are slight variations (subsets) of the aforementioned CO-II pattern. What differentiates these candidates is (i) polar vs. nonpolar character of the Fe double layer and (ii) symmetric vs. asymmetric Lu-layer displacement. The latter is closely associated with the phase shift across the ferroelectric domain wall in the superlattices[41]. We find that the single-domain type charge-ordering state (CO-FE) is ferroelectric (Fig. 4a). This is because the Lu-layer distorts in the same direction along $c$ and with the same down/up/up pattern throughout the material. On the other hand, the Lu-layer displacement is asymmetric in the doped-type state (CO-DOPED). Here, our calculations predict a spontaneous electron transfer from Fe²⁺ sites in the bilayer to Fe³⁺ sites in the LuFeO₃ layer (Fig. 4b). This leads to an Fe³⁺-rich bilayer, which increases magnetization in the LuFe₂O₄ slab—consistent with a larger coercive field and higher moment. It is the electron transfer that periodically reverses the Lu-layer distortion to create the asymmetric down/up/up and down/up/down pattern across the Fe bilayer. This changes the direction of electric polarization across each domain wall which acts to create a nonpolar Fe double layer and overall antiferroelectric state. As we shall see below, this is the state that corresponds most closely with experiment.

Figure 4c compares the dichroic spectra for the (3, 1) superlattice with our calculations. As discussed above, two different CO-II-type charge-ordering patterns were imposed in the simulations, with the goal of distinguishing between them. These include CO-FE and CO-DOPED (Fig. 4a, b). Overall, the experimental spectrum is in agreement with the CO-DOPED model. This means that the Fe double layer is nonpolar, and the Lu-layer displacement is asymmetric. Comparison reveals very similar results below $\approx 1.5$ eV for both states. The model predictions separate above this energy—similar to what we find for the case of bulk LuFe₂O₄ (Supplementary Fig. 7b). Spectral signatures that distinguish the CO-DOPED model include the minimum near 1.5 eV and sign change near 2 eV. The overall agreement becomes less quantitative at higher energies—possibly due to additional complexity in the charge-ordering pattern due to charged ferroelectric domain walls or reduced measurement sensitivity as the absorption coefficient rises (Fig. 2a). In any case, all of our calculations in Figs. 1c and 4c implement this particular charge-ordering pattern and are internally consistent. The CO-DOPED model is likely to apply to the higher-order superlattices ($m = 7$ and 9) due to the stronger Lu-layer distortion, although calculations cannot be performed at this time due to the extraordinary size of the unit cells. Our finding for the nonpolar CO-DOPED model is consistent with real space HAADF-STEM images as well (Fig. 1b)[20].

**Summary and outlook**. By combining optical absorption spectroscopy, high-field magnetic circular dichroism, and first-principles calculations, we unravel the microscopic origin of

high-temperature magnetism in the $(LuFeO_3)_m/(LuFe_2O_4)_1$ superlattices ($m = 3$, 7, and 9) and, at the same time, reveal the charge-ordering pattern in the $m = 3$ member of this family of multiferroic materials. Analysis of the site-specific coercivity vs. temperature curves, obtained from constant energy cuts of the dichroic spectra, demonstrates that bulk magnetism derives principally from the $LuFe_2O_4$ layers. Magnetism emanating from the $LuFe_2O_4$ layer becomes more robust as the $(3, 1) \rightarrow (7, 1) \rightarrow (9, 1)$ series progresses—a trend that correlates with increasing Lu-layer distortion. To understand this relationship more deeply, we extract the spectral signature of the interface for the $(LuFeO_3)_m/(LuFe_2O_4)_1$ series ($m = 3$, 7, and 9). While the overall contribution of spin-down channel excitations is persistent over the sequence, enhanced Lu-layer distortion at the interface increases the contribution of the $Fe^{2+} \rightarrow Fe^{3+}$ charge-transfer excitation in the spin-up channel. This amplifies $LuFe_2O_4$ layer magnetization and pinpoints the role of $Fe^{2+}$. Key to this discovery is the ability of magneto-optical spectroscopy to provide direct, microscopic, site-specific information about interface magnetism in a two-dimensional material with multiple magnetic centers. Comparison of the theoretically predicted magnetic circular dichroism with the experimental spectrum also establishes the nonpolar self-doped structure as the precise charge-ordering arrangement within the $LuFe_2O_4$ layer of the $(3, 1)$ film, thus resolving controversy regarding the many different isoenergetic charge states. In addition to introducing a remarkably powerful and versatile spectroscopic decomposition technique for revealing microscopic spin and charge character at the interface of a multiferroic superlattice with many different iron centers in a site-selective manner, this work provides a pathway to link bulk and interface properties in other engineered materials.

In our view, superlattice multiferroics remain a huge untapped frontier that is very likely to yield high-performance room temperature fully coupled multiferroics. $(LuFeO_3)_m/(LuFe_2O_4)_n$ superlattices are just the tip of the iceberg. Synthesizing these interface multiferroics is challenging. But understanding the inner workings of these interface materials is in its infancy and the spectroscopic decomposition method that we report is a powerful means to learn about how they tick with site-specific understanding directly at the interface. Analogous opportunities exist to exploit interface materials to enhance spintronics and photonics. As a result, there is broad utility in revealing interface dynamics well beyond the multiferroics community.

## Methods

**Film growth and structural characterization**. $(LuFeO_3)_m/(LuFe_2O_4)_1$ ($m = 3$, 7, and 9) thin films were grown using reactive-oxide molecular-beam epitaxy on (111) $(ZrO_2)_{0.905}(Y_2O_3)_{0.095}$ substrates. Lutetium and iron were evaporated from elemental sources and oxidized by a mixture of $\approx 2\%$ $O_3$ and $O_2$. The oxygen partial pressure was varied during the deposition to access the different iron valence states in $LuFe_2O_4$ ($Fe^{2.5+}$) and $LuFeO_3$ ($Fe^{3+}$). The $(LuFeO_3)_m/(LuFe_2O_4)_1$ ($m = 3$, 7, and 9) superlattices were grown as part of a full series of $(LuFeO_3)_m/(LuFe_2O_4)_n$ thin films to demonstrate consistent and reproducible trends in the ferroelectric and magnetic properties; characterization of the identical $(LuFeO_3)_3/(LuFe_2O_4)_1$ film presented here by x-ray diffraction and bulk SQUID magnetometry is presented in ref. [20]. The $(3, 1)$, $(7, 1)$, and $(9, 1)$ superlattices, as well as the two end member films of $LuFeO_3$ and $LuFe_2O_4$ were grown to a consistent number of iron layers to optimize the optical density and sensitivity for the transmission mode magnetic circular dichroism measurements. Cross-sectional TEM specimens were prepared using an FEI Strata 400 Focused Ion Beam with a final milling step of 2 keV to reduce surface damage. High-resolution HAADF-STEM images were acquired on an aberration-corrected 300 keV FEI Titan Themis with a probe convergence semiangle of 30 mrad. Information on the Fe valence and oxygen stoichiometry of these films is available in Supplementary Information.

**Optical spectroscopy**. We measured the $ab$-plane transmittance of the $(3, 1)$, $(7, 1)$, and $(9, 1)$ superlattices, the $LuFeO_3$ and $LuFe_2O_4$ end members, as well as the blank substrate using a $\lambda$-900 grating spectrometer covering the energy range from 1 to 6 eV. The linear absorption spectrum is calculated from measured transmittance as $\alpha(E) = -\frac{1}{d}\ln(T(E))$, where $T(E)$ is the measured transmittance as a

function of energy $E$ and $d$ is the sample thickness. The absolute absorption of $(3, 1)$, $(7, 1)$, and $(9, 1)$ superlattices, as well as the end members were determined by subtracting the response of the substrate. Because the optical density of the films was optimized for magnetic circular dichroism spectroscopy rather than linear absorption, the excitations are not as pronounced as in prior work[27,32]. An open-flow cryostat provided temperature control (4.2–300 K).

**Magnetic circular dichroism spectroscopy**. We measured the dichroic response of the superlattices ($m = 3$, 7, and 9), the $LuFeO_3$ and $LuFe_2O_4$ end members and the $(ZrO_2)_{0.905}(Y_2O_3)_{0.095}$ substrate between 0.8 and 2.8 eV. This is the energy window where our films transmit light. It is also the energy window where the most important excitations occur[27,32]. These experiments were performed at the National High Magnetic Field Laboratory using the 25 T split helix magnet[33] in Faraday geometry along with a 240 W Xe lamp and a 0.25 m monochromator. We measured the difference in transmittance between left- and right-circularly polarized (LCP and RCP) light at various magnetic fields, and converted the result to absorbance difference, as discussed in detail below. Thus, the dichroic spectrum is the difference in absorption between LCP and RCP. A chopper was employed to increase the signal to noise ratio at a constant frequency, followed by a linear polarizer that was set to 45°. A photoelastic modulator was placed after the linear polarizer to convert the linearly polarized light into left or right circular polarized light periodically at a constant time interval $\delta(t) = \lambda/4\sin(\omega t)$. We did not need to keep the phase information, so an optical fiber was used to collect the light and route it to the detector. All signals were separated by lock-in amplifiers. The field sequence was chosen based upon the needed resolution, always within the $+25$ $T \rightarrow 0^- T \rightarrow -25 T \rightarrow 0^+ T \rightarrow +25 T$ run pattern. The positive or negative sign of the magnetic field corresponds to the magnetic field direction and is parallel or antiparallel to the light propagation direction, respectively. The $0^-$ and $0^+$ are both zero-field data; the sign denotes the sweep direction. Moreover, a training loop with this pattern was performed before each data collection run. The phase of the lock-in was set at full field. Magnetic circular dichroism spectra were taken at several different temperatures—from ~18 to 157 K for the $(3, 1)$ and $(7, 1)$ superlattices. For the $(9, 1)$ superlattice, the temperature range was successfully increased to 218 K by adding an extra heater in the probe. Even so, we could not heat above this temperature.

**MCD data treatment**. In this work, we report the magnetic circular dichroism spectra in two different ways: as an absolute $\Delta\alpha_{MCD}$ for each superlattice or one that is normalized by the number of repeat units (which is just $\Delta\alpha_{MCD}/N$). Here, $N$ is the number of repeat units for the superlattices or end members. The latter rendering allows comparison of interface effects. Substrate correction to the magnetic circular dichroism spectrum is also important. As shown in Supplementary Fig. 2a, the MCD spectrum of the $(ZrO_2)_{0.905}(Y_2O_3)_{0.095}$ substrate is not zero because of the weak ferromagnetism induced from the defects[42]. We therefore subtracted it from the dichroic response of the superlattices to obtain the true $\Delta\alpha_{MCD}$ (or $\Delta\alpha_{MCD}$ per repeat unit). Details are available in Supplementary Information.

The magnetic circular dichroism spectra are obtained using the signal from the lock-in referenced to the photoelastic modulator divided by the signal from the lock-in referenced to the chopper. The chopper frequency is set to 217 Hz to improve the signal to noise ratio. There is, however, still some magnetic field-dependent background signal in the raw data (including the natural circular dichroism and the signal due to the drift of the probe), which dramatically affects the data quality when the dichroic signal from the sample is low—for instance in a nonmagnetic or antiferromagnetic material. To reveal the pure magnetic circular dichroism spectrum ($\Delta\alpha_{MCD}$), the field-induced background signal ($\Delta\alpha_{background}$) was subtracted from the total spectrum ($\Delta\alpha_{total}$) as: $\Delta\alpha_{MCD} = \Delta\alpha_{total} - \Delta\alpha_{background}$. At a given field $H$, we isolate $\Delta\alpha_{background}$ by averaging the positive and negative signals of the same field magnitude. This is because the $\Delta\alpha_{background}$ for both the positive and negative fields is only dependent on the intensity (and not the sign). In contrast, $\Delta\alpha_{MCD}$ depends on both the sign and the intensity of the magnetic field—making it an odd function. As a result, at the field of interest, the following relations should apply:

$$\Delta\alpha_{MCD} = \frac{1}{2} \times (\Delta\alpha_{+H} - \Delta\alpha_{-H}) \qquad (1)$$

$$\Delta\alpha_{background} = \frac{1}{2} \times (\Delta\alpha_{+H} + \Delta\alpha_{-H}), \qquad (2)$$

where $\Delta\alpha_{\pm H}$ is the raw MCD signal from the measurement at a positive or negative magnetic field $\mathbf{H}$, respectively. These equations indicate the pure MCD signal from the sample should be the average of the difference between the positive and negative fields. This method of analysis was applied to the $(3, 1)$, $(7, 1)$, and $(9, 1)$ superlattices, as well as to the spectra of the two end members.

**Extracting the coercive field and interface response from the dichroic spectra**. There are two aspects of the data treatment that deserve special mention. The first is the constant energy cuts of the MCD data. The second is the manner in which we extract the interface spectra. Constant energy cuts of $\Delta\alpha_{MCD}$ were used to reveal the behavior of specific Fe centers, and how the excitations of these centers

contribute to the overall magnetic response. By taking fixed energy cuts of the dichroic spectra over the full $+25\,\mathrm{T} \to 0^{-}\,\mathrm{T} \to -25\,\mathrm{T} \to 0^{+}\,\mathrm{T} \to +25\,\mathrm{T}$ data set, we can generate optical hysteresis loops corresponding to the excitation of interest. For instance, cuts at 1.33 eV probe the $Fe^{2+} \to Fe^{3+}$ charge-transfer excitation in the spin-down channel, and the hysteresis loop generated by these iron centers. We can extract a site-specific value of the coercive field from this type of optical hysteresis loop.

We also sought to isolate the interface response for each of the $(LuFeO_3)_m/(LuFe_2O_4)_1$ superlattices ($m = 3$, 7, and 9). This is important because high-temperature magnetism emanates from strain and rumpling at the interface. We could not, however, compare the measured MCD spectra directly because, even though each of the films were specifically designed to have a consistent number of Fe layers. This is because they have a different number of interfaces. To obviate this problem, we normalized the spectra by the number of repeat units. The MCD spectra per repeat unit is given by:

$$\Delta\alpha(E)_{\mathrm{per-repeat}} = \frac{\Delta\alpha(E)_{\mathrm{MCD}}}{N}, \qquad (3)$$

where $N$ is the number of repeat units. This quantity contains the information about the interface that we seek, but the response of the $LuFeO_3$ and $LuFe_2O_4$ layers has to be eliminated, in order to uncover it. To estimate the effect of the different $LuFeO_3$ and $LuFe_2O_4$ layers in the absence of the interfaces, we created a composite spectrum and subtracted this quantity from the measured MCD spectrum per repeat unit. We construct the composite spectrum of a hypothetical superlattice as:

$$\Delta\alpha_{\mathrm{composite}} = m \times \frac{\Delta\alpha_{LuFeO_3}}{N} + \frac{\Delta\alpha_{LuFe_2O_4}}{N}, \qquad (4)$$

where $m = 3$, 7, and 9. As mentioned above, $\Delta\alpha_{\mathrm{Interface}} = \Delta\alpha_{\mathrm{Measured}} - \Delta\alpha_{\mathrm{Composite}}$ on a "per repeat unit" basis. This process is discussed in detail in Supplementary Information and illustrated in Supplementary Fig. 3.

**First-principles electronic structure theory.** Density functional theory $+ U$ (DFT $+ U$) calculations were performed using the plane augmented wave method, as implemented in the Vienna Ab Initio Package (VASP), and selecting the Perdew–Burke–Ernzerhof form of exchange correlation functional. Lu $4f$ states were considered in the core and we set $U = 4.5$ eV and $J_H = 0.95$ eV for the Fe $3d$ states. DOS calculations were performed on the relaxed $(LuFeO_3)_3/(LuFe_2O_4)_1$ superlattice exhibiting a 2:1 ratio between $Fe^{3+}$ and $Fe^{2+}$ charges in each monolayer of the $LuFe_2O_4$ block. Specifically, in the $Fe^{3+}$–$Fe^{3+}$–$Fe^{2+}$ layer of the $LuFe_2O_4$ block, the $Fe^{3+}$ centers are antiparallel to each other; one of the $Fe^{3+}$ spins is up, and the other is down. The $Fe^{2+}$'s are in different layers and align ferromagnetically. The details of this structure which was found to be the lowest energy configuration among different charge orders were previously described in ref. [20]. The DOS were calculated with a $4 \times 4 \times 2$ $k$-point mesh and a kinetic energy cutoff of 500 eV. In order to probe the robustness of our results with respect to $U$, we also performed our calculations for a larger value, e.g., $U = 5.5$ eV. This introduces a global shift in the states above and below the Fermi level, but leaves the main features of the DOS unaltered. We also confirmed that this type of change in the value of $U$ does not impact our transition assignments.

**Absorption and magnetic circular dichroism spectra calculation based on different charge-ordering patterns.** The dichroic response can be modeled using the calculated matrix elements of the optical conductivity tensor as[36,43]:

$$\Delta\alpha_{\mathrm{MCD}} \approx \frac{d\omega}{2c} \Im(n_+ - n_-) \approx \frac{2\pi d}{c} \Im\left[\frac{\sigma_{xy}}{(1 + i\frac{4\pi}{\omega}\sigma_{xx})^{1/2}}\right]. \qquad (5)$$

Here, $n_\pm = (\epsilon_{xx} \pm \epsilon_{xy})^{1/2}$ is the refractive index of RCP or LCP light arising from the dielectric function $\epsilon$, $d$ is the film thickness, and $c$ is the speed of light. The theoretical predictions for both the parent compounds and the (3, 1) superlattice were made based on this equation. The dielectric functions were calculated using exact diagonalization as implemented in VASP. The theoretically predicted magnetic circular dichroism spectra were calculated based on the predicted DOS of different charge-ordering patterns in $LuFe_2O_4$ layers. The spin configuration considered in each case corresponds to the ferrimagnetic collinear arrangement of spins obtained from direct calculation of the magnetic ground state. They are mostly characterized by a ferromagnetic alignment of the $Fe^{2+}$ spins and an antiferromagnetic alignment of the $Fe^{3+}$ ones. The magnetic dichroism spectrum of the $LuFeO_3$ was computed considering the noncollinear $A2$ magnetic phase, which has been determined to be the magnetic ground state for this system and corresponds to a 120° angle in-plane (with a small tilt in the $z$-direction) arrangement of the spins within the Fe-monolayers in $LuFeO_3$[22]. Because $LuFeO_3$ (where the spins form a 120° noncollinear structure) is expected to provide a smaller contribution to $\Delta\alpha(E)$ than the $LuFe_2O_4$ layer (Supplementary Fig. 7a, b), we computed $\Delta\alpha(E)_{\mathrm{MCD}}$ of the two candidate charge-ordering models considering collinear spin structures in the $LuFeO_3$ layer.

## Data availability

Relevant data are available upon request from the corresponding author, J.L.M.

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

## Acknowledgements

Research at the University of Tennessee is supported by the U.S. Department of Energy, Office of Basic Energy Sciences, Materials Science Division under Award DE-FG02-01ER45885 (J.L.M.). Work at the National High Magnetic Field Laboratory is supported by the National Science Foundation through DMR-1644779 and DMR-1229217 (S.M.). Film growth and electron microscopy characterization is supported by the US Department of Energy, Office of Basic Energy Sciences, Division of Materials Sciences and Engineering, under Award No. DE-SC0002334. The Electron Microscopy Facilities at the Cornell Center for Materials are supported through the NSF MRSEC program (DMR-1719875). Research at the Tokyo Institute of Technology is supported by the Grant-in-Aid for Scientific Research 19K05246 from the Japan Society for the Promotion of Science (JSPS) and TSUBAME supercomputing facility. The use of the Maryland Advanced Research Computing Center (MARCC) is supported by the Platform for the Accelerated Realization, Analysis, and Discovery of Interface Materials (PARADIM): a National Science Foundation Materials Innovation Platform (under Cooperative Agreement No. DMR-1539918). Work at Berkeley is funded by the SRC ASCENT center under the SRC-JUMP program. We thank B. Donahoe and B. S. Holinsworth for useful discussions.

## Author contributions

This project was conceived by J.L.M. and S.F. The thin films were grown by J.M. and C.B. with advice from R.R. and D.G.S., and the STEM measurements were performed by M.H. and D.A.M. The MCD measurements were performed by S.F., K.A.S., S.M., and J.L.M., and data analysis was carried out by S.F., S.M., and J.L.M. The first-principles calculations were carried out by H.D. and A.R. with advice from C.J.F. The manuscript was written by S.F., H.D., K.A.S., A.R., R.R., D.G.S., and J.L.M. All authors discussed the data. All authors read the paper and commented on the manuscript.

## Competing interests

The authors declare no competing interests.
