## [Peer Review File · Nature Communications]

Reviewer #1 (Remarks to the Author):

The authors investigated Lutetium based superlattices formed by (LuFeO₃) and (LuFe₂O₄). In order to compare with theory, they chose (3,1) superlattices, i.e. 3 layers of LuFeO₃ and a single layer of LuFe₂O₄. To make further comparisons, the authors also prepared (7,1) and (9,1) superlattices as well. These superlattices were grown by using a reactive-oxide MBE system and the interface structures were imaged with atomic resolution TEM. First, the authors measured magnetic circular dichroism of the system using a high magnetic field. The measured magnetic circular dichroism spectra revealed distinct states at 1.5 eV, and 2.2 eV. The former was assigned as due to Fe²⁺ to Fe³⁺ charge transfer excitation. Then they measured the dichroic rotation at the energy of 1.33 eV, which belonged to the Fe²⁺ to Fe³⁺ transition by varying the temperature. The measured data showed temperature dependent hysteresis loops. This charge transfer excitation was originated from the LuFe₂O₄ layer, and comparison to the (7,1) and (9,1) superlattices, they further showed the minimum dependency of the charge transfer excitation at the spin-down channel. Unlike the spin down channel, the charge transfer at the spin-up channel was found to be critical. By means of first principle calculations, the authors proposed that the in-plane orbital degeneracy of the spin-up channel was the key for the robust magnetism in this material. The magnetism from LuFe₂O₄ become more prominent by increasing the LuFeO₃ layer, from (3,1) to (9,1). From this finding the authors conclude that the Lu layer distortion is responsible for the observed enhanced magnetism. This work appears to be solid, and an important finding to solve the interfacial magnetism. The site specificity here emerged from the in-plane orbital degeneracy in the spin-up channel, which is sensitive to the interfacial charge transfer. Clearly, the authors cleverly used it as an advantage to solve an important problem. However, at the current stage, it appears as more of a subject oriented approach to solve this particular superlattice. The authors should add a few sentences to discuss how this technique will be useful for general interfacial magnetism problems. If they highlight usefulness of this method more broadly, then this work will be suitable to publish in Nature Communications.

Reviewer #2 (Remarks to the Author):

In the present work, the authors revealed the origin of high-temperature magnetism in (LuFeO₃)_m/(LuFe₂O₄)₁ superlattices by combining optical absorption, magnetic circular dichroism and first-principles calculations. They found that the distortion of Lu layer at the interface selectively enhances the Fe²⁺ to Fe³⁺ charge transfer contribution in the spin-up channel and strengthens the exchange interactions. This is meaningful to study the complex oxide materials with multiple metal centers and strong entanglement. However, there are still some minor concerns and ambiguities in the present form of the manuscript.

- 1) It is better to provide the detailed crystal structure information of the end members of LuFeO₃ and LuFe₂O₄.
- 2) In the linear absorption spectrum, the authors only consider the contributions from three most important Fe-related excitations. How about the contributions from other elements (Lu and O) or defect states (oxygen vacancy and nonstoichiometry) in both LuFeO₃ and LuFe₂O₄? Please make some comments on this effect.
- 3) The valence of Fe is key for this study. How to convince that Fe^{+2.5} in LuFe₂O₄ and pure Fe⁺³ in LuFeO₃? What is the

contribution of Fe²⁺ in LuFeO₃?

4) Because the distortion of Lu layer is very important to the charge ordering and transfer at the LuFeO₃/LuFe₂O₄ interface, the displacement configurations of the Lu layers adjacent to the double Fe-layer should be considered carefully. For example, the displacements of upper Lu layer are symmetric or asymmetric with the bottom Lu layer, which may have obvious effect on the charge ordering. Please discuss about this.

Reviewer #3 (Remarks to the Author):

This manuscript presents results and analysis of spectroscopic measurements (optical circular dichroism) and DFT electronic structure calculations for superlattices of LuFeO₃ and LuFe₂O₄. Such superlattices exhibit both a spontaneous magnetization, originating from ferrimagnetic ordering, and a ferroelectric polarization close to room temperature (or perhaps above), and are therefore of great interest to the field of multiferroics.

The work seems to be a continuation of Ref. 1, where the multiferroic properties of these superlattices have first been reported by mostly the same group of authors. This, in fact, represents a certain problem of the present manuscript, which gives only a very brief introduction into the topic and does not properly introduce certain essential concepts. In essence, one first has to read Ref. 1 (including supplements etc.) to have any chance of understanding what is discussed in the present manuscript. In addition, I think that several of the conclusions are not really supported by the data, at least not in the way they are presented. There is a certain tendency to overstate what can be concluded from the obtained results (some specific examples are listed further below). I also find it very confusing that the conclusions are always presented before the actual data is discussed. The other way round would make the presentation much clearer.

In summary, I don't think that the manuscript is suitable for publication in its present form. Also, even if the presentation would be improved, I have doubts whether this work would merit the special publication criteria of Nature Communications.

I give some more detailed criticism in the following:

The first part of the manuscript is devoted to the analysis of the MCD spectra, which are decomposed into contributions stemming from different excitations, which are all located in the LuFe₂O₄ layers. The basis for this decomposition is the assignment of these excitations from the calculated DFT densities of states (DOS).

It would be interesting (and important!) to know what type of magnetic order was imposed in these calculations (I couldn't find this information anywhere, not even in Ref. 1), since the assignment of the different excitations of course depends on the local moment directions (i.e., what is locally spin up and spin down). For example, there seem to be different Fe³⁺ sites with opposite magnetic moments?

Furthermore, it is clear that the corresponding excitation energies depend sensitively on the value of U used in the calculations, so some more discussion of this would be desirable. How robust are the assignments and the resulting energy separations with respect to variations of U? The fact that the

same U value has been used in some other study is no justification for its universal applicability!

On the experimental side: can anything be concluded from the linear absorption spectrum shown in Fig. 2a? It appears to be essentially featureless and therefore does not give particularly strong support for the DFT transition assignments. Is it worth to include this at all?

Also, what is the purpose of presenting Eq. (1)? What follows from this equation in the context of this work? It also seems that several quantities appearing in this equation are not defined in the text.

Then, the separation of the spectra into individual contributions is quite interesting. It seems to work quite well, in spite of the uncertainties mentioned above. This leads to the conclusion that most of the magnetization in this system stems from the Fe double layer. While this appears quite reasonable, I think that strictly speaking one can just conclude that the global and the local coercivities are the same (Fig. 2f).

The second part of the paper then extends this analysis for multilayers with different number of LuFeO₃ layers. Here, again I find it quite difficult to follow what has actually been done. Since this is perhaps the most central part of the paper, I think that a more elaborate and clearer presentation would certainly be appropriate.

According to the authors, Fig. 3b "conclusively demonstrates that increasing magnetic moment of the LuFe₂O₄ layer emanates from rising .. density of states in the spin-up channel". This is totally unclear to me. In my understanding, Fig. 3b only shows that the MCD (and thus perhaps the corresponding magnetization) is enhanced in the (9,1) superlattice, and that this is mainly due to the excitations corresponding to the Fe²⁺-Fe³⁺ spin-up channel. I don't see a DOS comparison anywhere. It is also not clear to me how a magnetic moment follows from only the spin-up DOS. In my understanding it is mostly related to a difference between spin-up and spin-down DOS.

I also want to point out that even though Fig. 3b plots the MCD as function of the Lu layer distortion, both are in fact just a function of the superlattice periodicity. Therefore, what can be concluded is, that both enhanced MCD (and perhaps magnetization) and stronger Lu layer rumpling appear simultaneously in the superlattices with more LuFeO₃ layers, but not that the latter causes the former. An important difference!

This is followed by a discussion on how the Lu rumpling affects the orbital overlap (and thus likely the magnetic coupling) between the Fe cations in the double layer. However, this discussion remains rather superficial. For example, I cannot distinguish from Fig. 3e whether the nature of the distortion is mainly a tilt of the O bi-pyramids or whether also the Fe cations move along c?

I am also not sure what can be concluded from the projected DOS plots shown in Figs. 3f and g. The text states that the spin-up DOS is "enhanced". But compared to what? Only one case is shown in these Figs. Also, the claimed relation to the Lu layer distortion remains a mystery to me.

The final part is then somewhat disconnected from the previous focus of the paper. By comparing calculated MCD spectra for different charge order patterns (which, unfortunately, are neither defined nor described; maybe in Ref. 1?) it is concluded that the so-called CO-DOPED is the most likely one. If I understand correctly this case is not polar (maybe anti-polar?), so I wonder how this affects the room temperature multiferroic properties of this system.

Thus, as stated earlier, I think that the presentation and discussion of this manuscript needs significant improvements. Furthermore, instead of including overly bold statements, a more objective discussion on what can be concluded from this analysis would be preferable.

Referee 1 writes: The authors investigated Lutetium based superlattices formed by (LuFeO_3) and $(\text{LuFe}_2\text{O}_4)$. In order to compare with theory, they chose (3,1) superlattices, i.e. 3 layers of LuFeO_3 and a single layer of LuFe_2O_4 . To make further comparisons, the authors also prepared (7,1) and (9,1) superlattices as well. These superlattices were grown by using a reactive-oxide MBE system and the interface structures were imaged with atomic resolution TEM. First, the authors measured magnetic circular dichroism of the system using a high magnetic field. The measured magnetic circular dichroism spectra revealed distinct states at 1.5 eV, and 2.2 eV. The former was assigned as due to Fe^{2+} to Fe^{3+} charge transfer excitation. Then they measured the dichroic rotation at the energy of 1.33 eV, which belonged to the Fe^{2+} to Fe^{3+} transition by varying the temperature. The measured data showed temperature dependent hysteresis loops. This charge transfer excitation was originated from the LuFe_2O_4 layer, and comparison to the (7,1) and (9,1) superlattices, they further showed the minimum dependency of the charge transfer excitation at the spin-down channel. Unlike the spin down channel, the charge transfer at the spin-up channel was found to be critical. By means of first principle calculations, the authors proposed that the in-plane orbital degeneracy of the spin-up channel was the key for the robust magnetism in this material. The magnetism from LuFe_2O_4 become more prominent by increasing the LuFeO_3 layer, from (3,1) to (9,1). From this finding the authors conclude that the Lu layer distortion is responsible for the observed enhanced magnetism. This work appears to be solid, and an important finding to solve the interfacial magnetism. The site specificity here emerged from the in-plane orbital degeneracy in the spin-up channel, which is sensitive to the interfacial charge transfer. Clearly, the authors cleverly used it as an advantage to solve an important problem.

Our reply: Thank you for such a nice summary of our work.

Referee 1 writes: At the current stage, it appears as more of a subject oriented approach to solve this particular superlattice. The authors should add a few sentences to discuss how this technique will be useful for general interfacial magnetism problems. If they highlight usefulness of this method more broadly, then this work will be suitable to publish in Nature Communications.

Our reply: Good idea, thanks for the suggestion.

The last paragraph of the Summary and Outlook section now reads: “In our view interface multiferroics remain a huge untapped frontier that is very likely to yield high-performance room-temperature fully coupled multiferroics. $(\text{LuFeO}_3)_m/(\text{LuFe}_2\text{O}_4)_n$ superlattices are just the tip of the iceberg. Synthesizing these interface multiferroics is challenging. But understanding the inner workings of these interface materials is in its infancy and the breakthrough decomposition method that we report is a powerful means to learn about how they tick with site-specific understanding. Analogous opportunities exist to exploit interface materials to enhance spintronics and photonics. As a result, there is broad utility in revealing interface dynamics well beyond the multiferroics community.”

We also added the following text to the end of the Introduction: “In addition to introducing a remarkably powerful and versatile technique for revealing spin and charge character in interface materials, our work provides strategies for raising the magnetic ordering temperature in other engineered multiferroics.”

Thanks to these updates, our Communication demonstrates the utility of high magnetic fields and dichroic spectroscopy to probe and better understand magnetism and charge ordering in interface materials of all sorts and its publication in *Nature Communications* will help to disseminate the utility of this characterization method well beyond the multiferroic community. We very much appreciate Referee 1’s suggestion in this regard.

Reviewer 2:

Referee 2 writes: In the present work, the authors revealed the origin of high-temperature magnetism in $(\text{LuFeO}_3)_m/(\text{LuFe}_2\text{O}_4)_1$ superlattices by combining optical absorption, magnetic circular dichroism and first-principles calculations. They found that the distortion of Lu layer at the interface selectively enhances the Fe^{2+} to Fe^{3+} charge transfer contribution in the spin-up channel and strengthens the exchange interactions. This is meaningful to study the complex oxide materials with multiple metal centers and strong entanglement. However, there are still some minor concerns and ambiguities in the present form of the manuscript.

Our reply: Let's see what we can do to address these concerns.

Referee 2 writes: It is better to provide the detailed crystal structure information of the end members of LuFeO_3 and LuFe_2O_4 .

Our reply: We added the crystal structure of the two end members to the Supplementary Materials [Fig. S1] for easy comparison. Additional text is included as well.

Referee 2 writes: In the linear absorption spectrum, the authors only consider the contributions from three most important Fe-related excitations. How about the contributions from other elements (Lu and O) or defect states (oxygen vacancy and nonstoichiometry) in both LuFeO_3 and LuFe_2O_4 ? Please make some comments on this effect.

Our reply: The Referee is correct that we consider contributions from the three most important Fe-related excitations for the interpretation of the linear absorption spectrum and magnetic circular dichroism. This is due to the relative size of the matrix element for the different excitations. Let's take the excitations one at a time to see why this is so.

- The charge transfer excitations are quite strong because they involve Fe sites with two different charges. Because our samples are magnetic, charge transfer in the spin-up and spin-down channels are different. These excitations appear prominently in the linear absorption as well as the magnetic circular dichroism.
- The on-site Fe^{2+} *d-to-d* excitation is important for the dichroic response due to the large Fe^{2+} density of states in the spin-down channel [Fig. 1c]. This excitation is also evident in the optical absorption of the LuFe_2O_4 end member.
- On the other hand, the Fe^{3+} *d-to-d* excitation in the LuFe_2O_4 double layer is weak in the dichroic response due to antiferromagnetic behavior at the site. Moreover, the spin-up and spin-down aspects of the excitation cancel even this weak signature in the magnetic circular dichroism spectra, again reducing its overall importance [Fig. 1c]. This excitation is weak in the linear absorption spectra due to the LaPorte and spin-selection rules, although symmetry-breaking activates it slightly.
- The Fe^{3+} in the LuFeO_3 layers is antiferromagnetic, and as a result, the intensity of the on-site Fe^{3+} *d-to-d* excitation is fairly modest in the dichroic response [Fig. S2b].

This excitation is also weak in the linear absorption spectra due to the LaPorte and spin-selection rules, although symmetry-breaking activates it slightly.

- According to our calculations and previous literature, the Lu-related excitations appear at much higher energy (above 5 eV in the LuFe_2O_4 end member) - beyond our measured energy range. Also, Lu is non-magnetic, so Lu excitations hybridized with oxygen do not exhibit a magnetic circular dichroism signal.
- Referee 2 is correct that there are oxygen defects in samples of this type, particularly in the LuFe_2O_4 layer that hosts the magnetism. There is not a good lattice match for LuFeO_3 , so while the samples are phase pure and oriented, there are threading dislocations and similar types of structural defects. Similar dislocations are also present in the LuFeO_3 and LuFe_2O_4 parent compounds in approximately the same concentration. In our prior work, we report that the magnetic moment from these defects is at least an order of magnitude smaller than that of the host compound (Refs. 21 and 27 in the paper). We therefore conclude that the magnetic circular dichroic signal from these defects is extremely weak as well.

We agree that information on how we identify the most prominent excitations is extremely important and should be better highlighted. The first paragraph of the Results and Discussion section (on page 4) has been updated to include details on the most important assignments, and an additional section was added to Supplementary Materials (on page 4 and 5) to discuss these issues. We also updated the summary of excitations in Table I to include information on the relative intensities. Information on oxygen stoichiometry has been added to the Supplemental Materials (page 4). Thanks for the suggestion.

Referee 2 writes: The valence of Fe is key for this study. How to convince that $\text{Fe}^{+2.5}$ in LuFe_2O_4 and pure Fe^{3+} in LuFeO_3 ? What is the contribution of Fe^{2+} in LuFeO_3 ?

Our reply: In prior work, we studied the Fe valence in the LuFeO_3 and LuFe_2O_4 parent compounds. We grew single phase LuFeO_3 films by molecular-beam epitaxy and investigated the effect of stoichiometry on the magnetic properties (Ref. 21). Using our phase-pure samples, we demonstrated Fe^{3+} in LuFeO_3 samples by EELS. Annealing the samples in ozone post-synthesis did not change the observed valence or other properties, further suggesting that our synthesized samples were fully oxidized. We also note that excess Fe in these films tends to accumulate as Fe_3O_4 precipitates ($\text{Fe}^{2.67+}$) which are readily observed by magnetometry, AFM, and TEM imaging. The superlattice films used for magnetic circular dichroism spectroscopy, presented in the current manuscript, are free of these Fe_3O_4 inclusions as well.

We also explored the Fe valence in the LuFe_2O_4 end member by EELS (Ref. S8). While we observed an average $\text{Fe}^{+2.5}$ valence, we were unable to identify discrete Fe^{3+} and Fe^{2+} states from a charge ordering pattern. This could be due to instrumental limitations, for instance, performing a room temperature measurement during which the electron beam averages through a column of Fe atoms. This measurement was, however, in contrast to prior work (Ref. S9) which found discrete charge-ordering. Nevertheless, our samples are consistent with a bulk valence of $\text{Fe}^{2.5+}$.

The films measured and discussed in the current manuscript were synthesized by the same MBE techniques as in our prior work. Using EELS, we again confirmed the Fe^{3+} valence in LuFeO_3 in the superlattices. The iron valence in the LuFe_2O_4 layer is slightly different than the expected 2.5+ because of the spontaneous charge transfer from the Fe^{2+} site in the LuFe_2O_4 layer to the Fe^{3+} site in the LuFeO_3 layer [Fig. 5b]. (Ref. 43)

The referee is correct that our DFT models suggest that there should be a small amount of Fe^{2+} present in the LuFeO_3 layers in the superlattices as a result of the ferroelectric domain walls as shown in Fig 5b. Thus far, we have not been able to detect this accumulation of charge to screen the domain wall, in contrast to our ability to do so in a related hexagonal magnetite system (Ref. S11). The charge transferred to the domain wall should be approximately 0.1 e-, yielding a net valence of 2.9+ if it were to be accumulated on a single Fe site.

We can place bounds on the likely fraction of Fe^{2+} in the LuFeO_3 layers by examining the candidate domain wall structure in Fig. 5b. Assuming that 0.1 e- is transferred to only the middle Fe layer in LuFeO_3 - as our theoretical model suggests - we find a maximum Fe^{2+} fraction of 3%. Importantly, the overall dichroic signal from antiferromagnetic LuFeO_3 is much weaker than that of LuFe_2O_4 [Fig. S2b,c], so an additional signal from these Fe^{2+} centers in the LuFeO_3 layer would leave our conclusions unchanged. Our findings are therefore quite robust.

Referee 2 is correct that the Fe valence is key to this study. In order to provide this information to our readers, we added a section entitled “Fe valence in the $(\text{LuFeO}_3)_3/(\text{LuFe}_2\text{O}_4)_1$ superlattices” on pages 3 and 4 of the Supplementary Materials.

Referee 2 writes: Because the distortion of Lu layer is very important to the charge ordering and transfer at the $\text{LuFeO}_3/\text{LuFe}_2\text{O}_4$ interface, the displacement configurations of the Lu layers adjacent to the double Fe-layer should be considered carefully. For example, the displacements of upper Lu layer are symmetric or asymmetric with the bottom Lu layer, which may have obvious effect on the charge ordering. Please discuss about this.

Our reply: This is an excellent and extremely interesting question. The Referee is correct that the displacement pattern of Lu at the interface influences all aspects of the electronic structure including the charge ordering pattern and that different charge order patterns are connected with different interface distortions. This is the main point of the section entitled “Determining the charge ordering pattern in $(\text{LuFeO}_3)_3/(\text{LuFe}_2\text{O}_4)_1$ ” starting on page 10.

Here, we discuss how there are several nearly isoenergetic charge ordering patterns - and two, in particular, that are the lowest energy and indistinguishable by theory [Fig. 5a, b]. A simple energy minimization is clearly not enough to reveal the true ground state. We therefore simulate the magnetic circular dichroism spectrum using our calculated bands + optical conductivities [Eqn. 6] and, by comparing the measured and computed spectrum, we reveal that the CO-DOPED charge order pattern is the correct ground state. This is a state with the non-polar Fe bilayers in which the displacement pattern of the upper Lu

layer is asymmetric with respect to the bottom Lu layer [Fig. 5b]. All of the calculated results shown in Figs. 1c, and 5c implement this particular charge ordering pattern in an internally-consistent manner. The charge degrees of freedom have a much larger energy scale than the spin, so it makes sense that the magnetism depends upon the charge pattern. The symmetric vs asymmetric Lu-layer distortion is also strongly associated with the phase shift at the ferroelectric domain wall in the superlattices.

We agree with the Referee that the conditions under which our calculations are carried out is extremely important and they should be communicated more broadly. We therefore updated the caption and text surrounding Fig. 1c (page 23) to indicate that the calculations were carried out using the non-polar Fe bilayers charge ordering pattern and an asymmetric distortion of the Lu layer. These conditions are fully consistent with all experimental measurements.

We also added additional text in the section entitled “Determining the charge ordering pattern in $(\text{LuFeO}_3)_3/(\text{LuFe}_2\text{O}_4)_1$ ” (pages 10 and 11) to better address this issue. The changes include:

- Lines 255 and 256: “The challenge is that there are several nearly isoenergetic charge ordering candidates.”
- Lines 271 - 274: “What differentiates these candidates is (i) polar vs. non-polar character of the Fe double layer and (ii) symmetric vs. asymmetric Lu-layer displacement, which is closely associated with the phase shift at the ferroelectric domain wall in the superlattices.⁴³”
- Line 277: “the Lu-layer displacement is asymmetric in the doped-type state (CO-DOPED).”
- Lines 281 - 283: “This changes the direction of electric polarization across each domain wall which acts to create a non-polar Fe double-layer and overall antiferroelectric state. As we shall see below, this is the state that corresponds most closely with experiment.”
- Lines 287 and 288: “This means that the Fe double layer is non-polar, and the Lu-layer displacement is asymmetric.”
- Lines 294 and 295: “In any case, all of our calculations in Figs. 1c and 5c implement this particular charge ordering pattern and are internally consistent.”

We trust that together these modifications clarify the connection between the charge ordered ground state and the Lu-layer displacement. Thank you for the opportunity to update our text with these important clarifications.

Reviewer 3:

Referee 3 writes: This manuscript presents results and analysis of spectroscopic measurements (optical circular dichroism) and DFT electronic structure calculations for superlattices of LuFeO_3 and LuFe_2O_4 . Such superlattices exhibit both a spontaneous magnetization, originating from ferrimagnetic ordering, and a ferroelectric polarization close to room temperature (or perhaps above), and are therefore of great interest to the field of multiferroics.

Our reply: Yes, this is a very interesting family of materials. We especially like the opportunity to develop structure-property relations.

Referee 3 writes: The work seems to be a continuation of Ref. 1, where the multiferroic properties of these superlattices have first been reported by mostly the same group of authors. This, in fact, represents a certain problem of the present manuscript, which gives only a very brief introduction into the topic and does not properly introduce certain essential concepts. In essence, one first has to read Ref. 1 (including supplements etc.) to have any chance of understanding what is discussed in the present manuscript. In addition, I think that several of the conclusions are not really supported by the data, at least not in the way they are presented. There is a certain tendency to overstate what can be concluded from the obtained results (some specific examples are listed further below). I also find it very confusing that the conclusions are always presented before the actual data is discussed. The other way round would make the presentation much clearer.

Our reply: There are many issues here. Let's take them one at a time.

- Yes, Referee 3 is correct that many of the authors from Ref. 1 (Mundy et al, Nature (2016)) contributed to this work. The collaboration made the project a lot of fun.

The author list makes sense from the film growth point of view because the team of Schlom, Ramesh, Mundy, and Brooks are the only ones that know how to grow these superlattices with precision control of the Fe oxidation state, oxygen concentration, and interface quality. Without their collaboration, we could not be certain of top quality films. The same holds true for the microscopy group and the theory team from Cornell, although there are new additions to the author list there.

All of this aside, ours is a spectroscopy paper at heart. We utilize prior knowledge of how to grow the best films and how certain calculations are made, but we extend to magnetic circular dichroism spectroscopy in order to reveal the microscopic origin of the high temperature magnetic state and the charge ordering pattern of the Fe double layer in this family of materials. These topics can not be addressed by the film growers, the microscopy group, or the theory team in isolation.

Success required leadership from the spectroscopy side, and all of these authors (Fan, Smith, McGill, Musfeldt) are new to the collaboration. So rather than “representing a problem for the present manuscript”, we believe that the team is uniquely qualified to carry out this work. We hope that, upon consideration, the Referee will agree.

- Referee 3 also points out that we could do a better job introducing the topic and

providing background on essential concepts. At the same time, we endeavored to better articulate the open questions from the prior work in Ref. 1. These include the origin of high temperature magnetism and the charge ordering pattern in the Fe double layer. Several sentences were added to explain how and why we set about addressing them. The changes in the Introduction include:

- Line 58: “The crystal structures of end members are shown in Fig. S1.”
- Lines 60-62: “Such a superlattice has a higher magnetic ordering temperature than either of its end members due to interface effects¹. The microscopic nature of these interface effects and their connection to the more robust magnetism is highly under-explored.”
- Lines 64-67: “The symmetric vs. asymmetric displacement of the upper and lower Lu layers adjacent to LuFe_2O_4 is one characteristic that differentiates the polar vs. non-polar Fe double layer charge ordering candidates. Resolving these issues is crucial for determining the ground state.”
- Lines 72-74: “In addition to introducing a remarkably powerful and versatile technique for revealing spin and charge character in interface materials, our work provides strategies for raising the magnetic ordering temperature in other engineered multiferroics.”

We also added three sections in the Supplementary Materials detailing the crystal structures of the end members, Fe valence and defects in the $(\text{LuFeO}_3)_3/(\text{LuFe}_2\text{O}_4)_1$, and the assignments of the electronic excitations.

We agree that, taken together, these changes are quite useful and improve readability immensely.

- Finally, Referee 3 wonders whether the presentation should be reorganized to introduce the “data” before the “conclusions”. We puzzled over this suggestion for a long time and then, it finally occurred to us that the problem may be the following:

As an experimental spectroscopy group, we present the measurement, the analysis of the measurements, and the basic conclusions that can be gleaned from the experiment. Then we compare with the existing theoretical framework, verify or refute, and extend to allow the theorists to predict things that we can not unravel on our own. Then, where possible, we go back to test the computed result.

So yes, Referee 3 is correct that there is some recapitulation in the overall story line. He or she is also correct that there are certain cases where experimental findings come before theory. This is, however, because our “data” is the experimental data - not the theory - and where this occurs, our findings do not depend upon the calculations.

On the other hand, there does not seem to be a more logical way to present the story. This is not a theoretical paper at its heart, so it would be difficult to put the calculations first without any experimental motivation for them. Without the spectroscopic measurement to justify the calculation, they are just isolated computed results without clear theme and purpose tying them together.

That said, we appreciate the Referee’s concern about organization. We carefully examined our overall presentation to avoid leaping to conclusions that are not well justified by our measurements and made a few adjustments. We hope that the Referee agrees.

Referee 3 writes: In summary, I don’t think that the manuscript is suitable for publication in its present form. Also, even if the presentation would be improved, I have doubts whether this work would merit the special publication criteria of Nature Communications.

Our reply: In our view interface multiferroics remain a huge untapped frontier that is very likely to yield high-performance room-temperature fully coupled multiferroics. $(\text{LuFeO}_3)_m/(\text{LuFe}_2\text{O}_4)_n$ superlattices are just the tip of the iceberg. Synthesizing these interface multiferroics is challenging. But understanding the inner workings of these interface materials is in its infancy and the breakthrough method that we report is a powerful means to learn about how they tick with site-specific understanding. Analogous opportunities exist to exploit interface materials to enhance spintronics and photonics. As a result, there is broad utility in revealing interface dynamics well beyond the multiferroics community. Thanks to the feedback from Referee 3, our Communication demonstrates the utility of high magnetic fields and dichroic spectroscopy to probe and better understand magnetism and charge ordering in interface materials of all sorts and its publication in *Nature Communications* will help to disseminate the utility of this characterization method well beyond the multiferroic community. We appreciate Referee 3’s feedback in this regard.

Referee 3 writes: The first part of the manuscript is devoted to the analysis of the MCD spectra, which are decomposed into contributions stemming from different excitations, which are all located in the LuFe_2O_4 layers. The basis for this decomposition is the assignment of these excitations from the calculated DFT densities of states (DOS).

It would be interesting (and important!) to know what type of magnetic order was imposed in these calculations (I couldn’t find this information anywhere, not even in Ref. 1), since the assignment of the different excitations of course depends on the local moment directions (i.e., what is locally spin up and spin down). For example, there seem to be different Fe^{3+} sites with opposite magnetic moments?

Our reply: Yes, Referee 3 is correct that the different types of magnetic order can profoundly affect the electronic structure of the $(\text{LuFeO}_3)_m/(\text{LuFe}_2\text{O}_4)_1$ superlattices, and we appreciate the opportunity to clarify this point. The relevant text in the Methods section in lines 463-473 now reads “The spin configuration considered in each case corresponds to the ferrimagnetic collinear arrangement of spins obtained from direct calculation of the magnetic ground state. They are mostly characterized by a ferromagnetic alignment of the Fe^{2+} spins and an antiferromagnetic alignment of the Fe^{3+} ones. The magnetic dichroism spectrum of the LuFeO_3 was computed considering the non-collinear $A2$ magnetic phase, which has been determined to be the magnetic ground state for this system and corresponds to a 120° angle in-plane (with a small tilt in the z -direction) arrangement of the spins within the Fe-monolayers in LuFeO_3 (Ref. 22). Because LuFeO_3 (where the spins form a 120° non-collinear structure) is expected to provide a smaller contribution to $\Delta\alpha(E)$ than the LuFe_2O_4

layer [Fig. S6(a, b)], we computed $\Delta\alpha(E)_{MCD}$ of the two candidate charge-ordering models considering collinear spin structures in the LuFeO_3 layer.”

Referee 3 writes: Furthermore, it is clear that the corresponding excitation energies depend sensitively on the value of U used in the calculations, so some more discussion of this would be desirable. How robust are the assignments and the resulting energy separations with respect to variations of U ? The fact that the same U value has been used in some other study is no justification for its universal applicability!

Our reply: We agree that prior use of a U value does not guarantee general applicability. For this reason, we probed the robustness of our results by repeating our calculations for different U values (for instance 4.5 and 5.5 eV). As expected, there is a rigid shift of the states, but the main characteristics of the density of states remain unchanged. We have added this information into the Methods Section and lines 449-453 now read: “In order to probe the robustness of our results with respect to U , we also performed our calculations for a larger value, *e.g.*, $U=5.5$ eV. This introduces a global shift in the states above and below the Fermi level but leaves the main features of the DOS unaltered. We also confirmed that this type of change in the value of U does not impact our transition assignments. ”

Referee 3 writes: On the experimental side: can anything be concluded from the linear absorption spectrum shown in Fig. 2a? It appears to be essentially featureless and therefore does not give particularly strong support for the DFT transition assignments. Is it worth to include this at all?

Our reply: Referee 3 is correct that features in the linear absorption spectra are broad and smooth. This is because the optical density of our superlattice films was optimized for the magnetic circular dichroism measurements - not for the linear absorption.

This does not mean, however, that the linear absorption spectrum is useless.

In fact, it is necessary to include the linear absorption in the main part of our paper because, although the features are not prominent, we can still see slope changes that are linked to features in the magnetic circular dichroism. The first-derivative relations between the linear absorption and magnetic circular dichroism are given by Eqn. (1). Any inflection point in the linear absorption spectra will be dramatically magnified in the dichroic response.

Referee 3 has a good point when he/she notes that text describing this relationship near Eqn. 1 should be improved. It now reads: “Equation 1 nicely highlights the first-derivative relationship between the linear absorption and magnetic circular dichroism. Any inflection point in the linear absorption spectra will be amplified in the dichroic response.”

Referee 3 writes: Also, what is the purpose of presenting Eq. (1)? What follows from this equation in the context of this work? It also seems that several quantities appearing in this equation are not defined in the text.

Our reply: Eqn. (1) is incredibly useful for articulating the relationship between energy,

electronic structure, and magnetism. Two points are pertinent:

- Equation 1 directly correlates the measured dichroic spectrum to the projection of the magnetic moment along the light propagation direction. Thanks to this connection, we can assign particular excitations in the dichroic spectra thus allowing for a site-specific analysis of the magnetism. This is how we separate the response of the multiple magnetic centers and by so doing unravel the microscopic origin of the high temperature magnetism in the $(\text{LuFeO}_3)_m/(\text{LuFe}_2\text{O}_4)_1$ superlattices.
- At the same time, Eqn. 1 reveals a first-derivative relationship between the magnetic circular dichroism spectrum and the linear absorption. The consequence of this relationship is that any slope change in the linear absorption will be magnified in the dichroic response. We can then use the assignments from the linear absorption to assign the features in the magnetic circular dichroism spectra. This allows us to link the magnetic properties to the specific density of states - even to specific bands. As a result, we can perform a site-specific analysis of the magnetic behavior of each individual magnetic center.

We updated the text associated with Eqn. (1) to better highlight its usefulness. All parameters are defined as well.

Referee 3 writes: Then, the separation of the spectra into individual contributions is quite interesting. It seems to work quite well, in spite of the uncertainties mentioned above. This leads to the conclusion that most of the magnetization in this system stems from the Fe double layer. While this appears quite reasonable, I think that strictly speaking one can just conclude that the global and the local coercivities are the same (Fig. 2f).

Our reply: We appreciate Referee 3's encouragement regarding our spectral decomposition technique. The key aspect of this process is the realization that the optical hysteresis loops correspond to a particular energy and thus a specific Fe site. This allows us to directly address all of the different Fe-related excitations as indicated in Fig. 1(c) and Table I. Referee 3 goes on to wonder why we do not conclude that the global and local coercivities are the same. Yes, the global and local coercivities are nearly identical. The difference is due to small contributions from Fe^{3+} centers [Fig. S2e].

The relevant text in lines 161 - 165 now reads: "The trend in H_c is similar for all Fe centres, and the extracted coercive fields are in excellent agreement with bulk magnetization^{1,21}. This demonstrates that a significant portion of the magnetism in the (3, 1) superlattice originates from the LuFe_2O_4 layer. In other words, the global coercive field is approximately equal to the local coercive field in the LuFe_2O_4 layer."

Referee 3 writes: The second part of the paper then extends this analysis for multilayers with different number of LuFeO_3 layers. Here, again I find it quite difficult to follow what has actually been done. Since this is perhaps the most central part of the paper, I think that a more elaborate and clearer presentation would certainly be appropriate.

Our reply: Yes, it was challenging to write this portion of the paper because it's easy to provide too many methodological details (which tends to take away from the science findings) or too few details (which makes the overall rational difficult to follow). As we are sure Referee 3 will agree, it's best to aim for middle ground. In this spirit, we suspect that a better explanation of the "composite spectra" and how it is used to unravel the interface spectra is in order - not just in the Methods and Supplement (where it already exists) but in the main text.

To this end, we added several sentences on page 7 and 8. they reads: "We began by normalizing the magnetic circular dichroism spectra on a "per repeat unit" basis by taking the raw dichroic signal and dividing by the number of repeat layers [Eqn. (4)]. Next, we used the normalized spectra of two end members to construct a "composite spectrum" for each superlattice. The composite response is simply the dichroic signal generated from combining the per repeat unit end member spectra based upon the composition as given by m and n [Eqn. (5)]. We extract the interface spectrum of each film by subtracting the composite response from the measured spectrum on a "per repeat unit" basis. A detailed discussion of this procedure is available in the Methods section and Supplementary Materials."

We appreciate the suggestion and agree that it's important to get this balance correct.

Referee 3 writes: According to the authors, Fig. 3b "conclusively demonstrates that increasing magnetic moment of the LuFe_2O_4 layer emanates from rising .. density of states in the spin-up channel". This is totally unclear to me. In my understanding, Fig. 3b only shows that the MCD (and thus perhaps the corresponding magnetization) is enhanced in the (9,1) superlattice, and that this is mainly due to the excitations corresponding to the Fe^{2+} - Fe^{3+} spin-up channel. I don't see a DOS comparison anywhere. It is also not clear to me how a magnetic moment follows from only the spin-up DOS. In my understanding it is mostly related to a difference between spin-up and spin-down DOS.

Our reply: Referee 3 is correct that the MCD (and magnetization) is enhanced in the (9,1) superlattice and that this is mainly due to excitations corresponding to $\text{Fe}^{2+} \rightarrow \text{Fe}^{3+}$ in the spin-up channel. We emphasize that these findings are experimental in nature and rely only upon Figs. 3a,c.

Here's how it works. We can see from Fig. 3c that only the remnant magnetization emanating from the spin-up channel charge transfer excitation rises with increasing number of LuFeO_3 layers (or the Lu-layer distortion), whereas the magnetization emanating from the other two excitations in the spin-down channel is nearly unchanged. This means that increasing the number of LuFeO_3 layers (or the Lu-layer distortion) selectively enhances the magnetic contribution from the spin-up channel charge-transfer excitation.

It is possible that this trend may be associated with the self-doped charge-ordering pattern (Fig. 5b). As discussed in the text, the self-doped type charge-ordering state results from a spontaneous charge-transfer between the Fe^{2+} site in the double layer to the Fe^{3+} site in the LuFeO_3 layer. This process is anticipated to be more stable in higher order superlattices

($m = 7$ and 9), although exact calculations can not be performed at this time due to the size of the cell. We speculate that as additional LuFeO_3 layers are added, more electrons are transferred to the LuFeO_3 reservoir. Such a process would raise the Fe^{3+} concentration in the bilayers. Based on the fact that spin-up channel charge-transfer in the Fe bilayer becomes more important with increasing m [Fig. 3c], it is possible that the enhanced Fe^{3+} density in the bilayer resides in the spin-up channel. Such behavior has the potential to exaggerate the difference between the spin-up and spin down channel excitations and thus raise the total magnetization of the LuFe_2O_4 layer.

Referee 3 is correct that our long-term objective is to unravel these processes in the $(\text{LuFeO}_3)_m/(\text{LuFe}_2\text{O}_4)_n$ superlattices. Unfortunately, the relationships are complex and not amenable to obvious solutions. We appreciate that the Referee brought up this important point which encouraged us to think further about the mechanism as postulated above. While we do not feel comfortable including this type of discussion in our manuscript, it will certainly be a point of continuing work. One such collaboration that is already on-going includes an effort to examine domain wall stability as well as the relationship of that stability to charge at the wall. Once the covid situation abates, we are also planning linear magnetic dichroism work which may further reveal unique charge effects in the (3, 1), (7, 1), and (9, 1) series.

Referee 3 writes: I also want to point out that even though Fig. 3b plots the MCD as function of the Lu layer distortion, both are in fact just a function of the superlattice periodicity. Therefore, what can be concluded is, that both enhanced MCD (and perhaps magnetization) and stronger Lu layer rumpling appear simultaneously in the superlattices with more LuFeO_3 layers, but not that the latter causes the former. An important difference!

Our reply: Good suggestion. We replotted Fig. 3c as “ $\Delta\alpha_{MCD, rem.}$ and Lu-layer distortion vs. number of LuFeO_3 layers” to better illustrate the correlation between Lu-layer distortion, $\Delta\alpha_{MCD, rem.}$, and superlattice periodicity - without implying causality. The figure caption on page 25 and the associated text in the structure-property section have been updated as well. These changes include:

- Line 201: Replacing the phrase “Lu-layer distortion” with “superlattice periodicity”.
- Lines 202-205: “Figure 3c displays $\Delta\alpha_{MCD, rem.}$ for the different Fe-related excitations as a function of the number of LuFeO_3 layers. Because superlattice periodicity and the Lu-layer distortion are correlated, there is a relationship between $\Delta\alpha_{MCD, rem.}$ and the Lu-layer distortion as well.”
- Lines 208 and 209: Replacing the phrase “Lu-layer distortion” with “the number of LuFeO_3 layers (and the Lu-layer distortion)”.
- Lines 209 - 213: “This behaviour demonstrates that the increasing magnetic moment in the LuFe_2O_4 layer emanates from rising Fe^{2+} and Fe^{3+} density of states in the spin-up channel and, at the same time, it provides a microscopic explanation for how high-temperature magnetism in these superlattices derives from Lu-layer distortion as well as the more growth-oriented parameter of superlattice periodicity.”

- Page 25: In the figure caption, we replace the sentence “Remanent magnetic circular dichroism for different types of Fe-related excitations vs. Lu-layer distortion” to “Remanent magnetic circular dichroism for different types of Fe-related excitations and Lu-layer distortion vs. the number of LuFeO_3 layers (m).”

We agree that these changes clarify the situation.

Referee 3 writes: This is followed by a discussion on how the Lu rumpling affects the orbital overlap (and thus likely the magnetic coupling) between the Fe cations in the double layer. However, this discussion remains rather superficial. For example, I cannot distinguish from Fig. 3e whether the nature of the distortion is mainly a tilt of the O bi-pyramids or whether also the Fe cations move along c ?

Our reply: The discussion of interface design rules is indeed more qualitative. Our goal is to use the general predictions of the Goodenough-Kanamori-Anderson rules (on how local structure distortions modify orbital overlap and thus the exchange interaction) to explain the trends in the (3, 1), (7, 1), and (9, 1) series. In addition to using these rules to analyze how orbital overlap changes in response to Lu-layer distortion, we sought to gain insight into the magnetism and magnetoelectric coupling. Here, it’s important to realize that the rotation and elongation of the Fe^{3+} polyhedra have been predicted by theory and observed in the STEM images. It is then well-within the Goodenough-Kanamori-Anderson rules to claim that rotations of the Fe^{3+} polyhedra modifies orbital overlap in the Fe^{3+} -O- Fe^{3+} linkages, affecting the superexchange interaction and raising the magnetic Curie temperature.

The challenge, of course, is to diagram these effects.

We modified Fig. 4c to clearly show that there is a rotation and elongation of the bipyramids but no Fe off-centering along c . The text surrounding this figure was also updated for clarity. Now it reads (Lines 250 and 251): “This level of off-centering is negligible and does not contribute to the properties of these materials.”

We also modified the sentence at lines 235 and 236, now it reads: “The Goodenough-Kanamori rules govern how local structure distortions modify orbital overlap and exchange interactions³⁹.”

Referee 3 writes: I am also not sure what can be concluded from the projected DOS plots shown in Figs. 3f and g. The text states that the spin-up DOS is ”enhanced”. But compared to what? Only one case is shown in these Figs. Also, the claimed relation to the Lu layer distortion remains a mystery to me.

Our reply: We apologize for the confusion. We were trying to use these highly localized calculations to emphasize how the density of states is correlated with magnetization, but the $(\text{LuFeO}_3)_m/\text{LuFe}_2\text{O}_4$ superlattices are very complex systems, and it is clear that this super-simplistic discussion is not helping. To better clarify the situation, we split the figure into two parts. The new Fig. 3 shows the extracted interface properties which are wholly experimental, and the new Fig. 4 displays the Goodenough-Kanamori analysis of these

processes. Panels f and g from the old figure were removed to avoid confusion.

Referee 3 writes: The final part is then somewhat disconnected from the previous focus of the paper. By comparing calculated MCD spectra for different charge order patterns (which, unfortunately, are neither defined nor described; maybe in Ref. 1?) it is concluded that the so-called CO-DOPED is the most likely one. If I understand correctly this case is not polar (maybe anti-polar?), so I wonder how this affects the room temperature multiferroic properties of this system.

Our reply: Although our work focuses on understanding the microscopic origin of high temperature magnetism in the $(\text{LuFeO}_3)_m/(\text{LuFe}_2\text{O}_4)_1$ family of materials, we can not model these processes without a detailed understanding of the ground electronic state. This is because in addition to establishing charge ordering pattern, the ground state determines whether the Lu layer is distorted (or not) as well as how much it is distorted. As mentioned in the text, there are two nearly isoenergetic charge ordered ground state candidates that can not be distinguished by theory. It is possible that a clearer ground state might emerge if calculations could be performed on the (7, 1) and (9, 1) superlattices because we know from experimental observation that the Lu distortion increases in the series, but the overall size of the system precludes that approach at this time. We therefore had to take another approach.

Since all of our calculations to understand the magnetic properties are dependent upon the electronic ground state, we sought a very stringent test to unambiguously determine the electronic ground state. It turns out that simulating the magnetic circular dichroism spectrum of the (3, 1) superlattice provides such a test. We find that the state with the non-polar Fe double layer best matches the spectroscopic data.

Armed with the correct electronic ground state, we proceeded to compute all of the other theoretical quantities shown in our manuscript. Determination of the charge ordered ground state is thus intimately connected to the other parts of our work. As Referee 3 points out, the electronic ground state is precisely what drives the room temperature multiferroic properties of this system.

That said, we agree that clarification is needed - both in the section entitled “Determining the charge ordering pattern in $(\text{LuFeO}_3)_3/(\text{LuFe}_2\text{O}_4)_1$ ” as well as earlier in the text. To that end, we made several modifications. They include:

- Lines 64 - 67: “The symmetric vs. asymmetric displacement of the upper and lower Lu layers adjacent to LuFe_2O_4 is one characteristic that differentiates the polar vs. non-polar Fe double layer charge ordering candidates. Resolving these issues is crucial for determining the ground state.”
- Lines 255 and 256: “The challenge is that there are several nearly isoenergetic charge ordering candidates.”
- Lines 271 - 274: “What differentiates these candidates is (i) polar vs. non-polar character of the Fe double layer and (ii) symmetric vs. asymmetric Lu-layer displacement.

The latter is closely associated with the phase shift at the ferroelectric domain wall in the superlattices⁴³.”

- Line 277: “the Lu-layer displacement is asymmetric in the doped-type state (CO-DOPED).”
- Lines 281 - 283: “This changes the direction of electric polarization across each domain wall which acts to create a non-polar Fe double-layer and overall antiferroelectric state. As we shall see below, this is the state that corresponds most closely with experiment.”
- Lines 287 and 288: “This means that the Fe double layer is non-polar, and the Lu-layer displacement is asymmetric.”
- Lines 294 and 295: “In any case, all of our calculations in Figs. 1c and 5c implement this particular charge ordering pattern and are internally consistent.”

All of the calculated results shown in Figs. 1c, and 5c implement this particular charge ordering pattern in an internally-consistent manner. The charge degrees of freedom have a much larger energy scale than the spin, so it makes sense that the magnetism depends upon the charge pattern.

Referee 3 writes: Thus, as stated earlier, I think that the presentation and discussion of this manuscript needs significant improvements. Furthermore, instead of including overly bold statements, a more objective discussion on what can be concluded from this analysis would be preferable.

Our reply: We sincerely appreciate Referee 3’s time and real effort to offer constructive, interesting, and actionable feedback on our manuscript. Each suggestion was carefully and fully implemented to capture both the letter and intended spirit. There is no question that these modifications change the character of our paper. The introduction and background are more complete, the electronic excitations are better identified, and the charge ordering section is more strongly integrated with the discussion on high temperature magnetism. We hope that Referee 3 can see the difference too. Again, we appreciate his/her efforts.

Reviewer #1 (Remarks to the Author):

The authors have responded this reviewer request satisfactorily. I recommend its publication in Nature Communications.

Reviewer #2 (Remarks to the Author):

I think it is acceptable now.

Reviewer #3 (Remarks to the Author):

In the revised version the authors have improved the manuscript and in particular the clarity and consistency of its presentation, also in response to the critique by the other referees. However, there are still a number of issues where I feel that crucial information is missing or certain things are not properly introduced.

1. The "Lu displacements" should be properly introduced and explained. After devoting quite some time to both the original and the revised manuscript, I believe that I now understand that this relates to the "up-up-down" etc. displacements along c , indicated in Fig. 5 as net dipole moments (red and blue arrows). Or maybe not? This seems somewhat central to the whole study and I don't understand why it is not properly introduced. The same holds for the relationship of the corresponding distortion amplitude and the number of LuFeO_3 layers.

2. In the revised intro it is stated that "our work provides a strategy for raising the magnetic ordering temperature in other engineered multiferroics". It is unclear what that strategy is. Using more LuFeO_3 layers? That seems rather restricted to these specific superlattices.

3. I still don't see the point of including Eq. (1). While it is certainly intriguing that this relates the energy derivative of the total absorption to the MCD, I don't see a big benefit in the specific context of this work. Both total adsorption and MCD are measured and can be compared and related to each other without using Eq. (1). Also the mentioned correlation to the joint density of states does not seem to be used in the present context. In fact, it appears that the equation is rather approximate, since clearly the measured MCD spectrum changes sign while the derivative of the total absorption appears to be always positive. But of course it's up to the authors what to include in their paper...

4. As stated in my original report, the relation between the enhancement of the spin-up CT contribution to the MCD and the enhanced magnetic moment of the LuFe_2O_4 layer is far from obvious. The ideas that the authors outline in their response seem promising to better understand this relationship, but as the authors write themselves, these ideas are still somewhat premature and perhaps not fully conclusive. I agree that it might not be appropriate to include them in the present paper. However, then I also think that statements in the text such as "the magnetic moment emanating from the spin-up channel $\text{Fe}^{2+} \rightarrow \text{Fe}^{3+}$ charge-transfer excitation" (lines 192/193) or "the increasing magnetic moment in the LuFe_2O_4 layer emanates from rising Fe^{2+} and Fe^{3+} density of states in the spin-up channel" (lines 209/210) are not very instructive. Also, the statement in lines 213-215, that one can understand this relation by considering the charge-ordered state in greater detail, is not very useful if this relationship is not further discussed in the present paper.

5. Regarding the discussion around Fig. 4, how the Lu distortion affects the orbital overlap and magnetic coupling, I still think that the discussion is rather superficial and not overly instructive. That the distortion changes the orbital overlap and that this will affect the magnetic coupling appears to be nearly trivial. However, why it actually seems to *enhance* the magnetic coupling (and thus the magnetic ordering temperature) remains unclear.

6. The discussion of the different charge ordering pattern is still somewhat confusing. First, charge order patterns termed CO-I and CO-II are mentioned, then CO-FE and CO-DOPED are discussed. After having a look at the supplementary, it seems to me that CO-I and CO-II correspond to the LuFe_2O_4 parent compound, while the other two correspond to the superlattices (?). This should be clearly stated.

7. I appreciate the clarifications of the magnetic state used in the calculations. However one thing is still unclear to me. What does "antiferromagnetic alignment of the Fe^{3+} " mean? Are they anti-parallel with respect to the Fe^{2+} moments or antiparallel with respect to each other (i.e, Fe^{2+} and half of the Fe^{3+} are "up", and the rest of the Fe^{3+} "down")?

In summary, I think that with a few more clarifications the manuscript is in principle suitable for publication. I also think it provides clear progress in understanding this specific class of superlattices, which are indeed important examples of "engineered multiferroics". As mentioned, I don't see any clear strategies for improving the properties of other multiferroic systems following from this work. I assume that the method of separating the different contributions to the MCD can also be useful for other systems, but whether it's a "breakthrough method" I don't know. It certainly depends on the specific electronic structure whether it is possible to have a clear energy separation between the different contributions. I am therefore not sure whether the manuscript is specifically suited for Nature Communications, but I hope that the editor will be able to decide this based on the given information.

Reviewer 1

Referee 1 writes: *The authors have responded this reviewer request satisfactorily. I recommend its publication in Nature Communications.*

Our reply: Thank you for your suggestion. We agree that it broadens the impact.

Reviewer 2:

Referee 2 writes: *I think it is acceptable now.*

Our reply: Thanks again for the useful feedback regarding the Fe valence and oxygen stoichiometry. We appreciate it.

Reviewer 3:

Referee 3 writes: *In the revised version the authors have improved the manuscript and in particular the clarity and consistency of its presentation, also in response to the critique by the other referees. However, there are still a number of issues where I feel that crucial information is missing or certain things are not properly introduced.*

Our reply: Thank you for noticing our improvements and for the additional suggestions. We are happy to address any issues that interfere with clear communication and as always, appreciate the ability to do so.

Referee 3 writes: *The "Lu displacements" should be properly introduced and explained. After devoting quite some time to both the original and the revised manuscript, I believe that I now understand that this relates to the "up-up-down" etc. displacements along c , indicated in Fig. 5 as net dipole moments (red and blue arrows). Or maybe not? This seems somewhat central to the whole study and I don't understand why it is not properly introduced. The same holds for the relationship of the corresponding distortion amplitude and the number of LuFeO_3 layers.*

Our reply: Yes, Referee 3 is correct that the connection between Lu-layer displacement, the size and pattern of displacements along c , and the number of LuFeO_3 layers is extremely important to our discussion. That is why we summarize the relationships between these quantities in Fig. 1. We agree, however, that this can be improved. We modified Fig. 1 so that the the Lu-layer distortion d is shown in red, and we updated the caption and text to be more clear in this regard. We also updated the caption of Fig. 4 so that the relationship between the distortion pattern and the red + blue net dipole moments is apparent.

Referee 3 writes: *In the revised intro it is stated that "our work provides a strategy for raising the magnetic ordering temperature in other engineered multiferroics". It is unclear*

what that strategy is. Using more LuFeO₃ layers? That seems rather restricted to these specific superlattices.

Our reply: No, of course not. Merely adding more LuFeO₃ layers is dull in the extreme, although this is the kind of thing that growth teams do as a comprehensive part of their development work. As spectroscopists, we are more curious about *how* adding more layers influences properties - not just simple bulk properties like magnetization and coercive field but hidden properties such as the relative importance of certain types of excitations and density of states at the interface. Recently, we have been thinking about materials design strategies that bring theory into the picture much earlier in the process and how to provide some of the more complex interface properties needed to benchmark electronic structure predictions of heterostructured materials. In the long term, this strategy is likely to speed the search for strongly coupled room temperature multiferroics. But the idea seems to have created confusion, so we deleted the sentence.

The final sentences of the Introduction now read: “In addition to introducing a remarkably powerful and versatile technique for extracting spin and charge character at the interface in a homologous series of multiferroic heterostructures, our work opens the door to similar approaches in other engineered materials as well as opportunities for the development of structure-property relationships and interface descriptors, potentially advancing a number of allied fields including spintronics and photonics.”

Referee 3 writes: *I still don't see the point of including Eq. (1). While it is certainly intriguing that this relates the energy derivative of the total absorption to the MCD, I don't see a big benefit in the specific context of this work. Both total adsorption and MCD are measured and can be compared and related to each other without using Eq. (1). Also the mentioned correlation to the joint density of states does not seem to be used in the present context. In fact, it appears that the equation is rather approximate, since clearly the measured MCD spectrum changes sign while the derivative of the total absorption appears to be always positive. But of course it's up to the authors what to include in their paper...*

Our reply: We removed Eqn. 1 and the associated text.

Referee 3 writes: *As stated in my original report, the relation between the enhancement of the spin-up CT contribution to the MCD and the enhanced magnetic moment of the LuFe₂O₄ layer is far from obvious. The ideas that the authors outline in their response seem promising to better understand this relationship, but as the authors write themselves, these ideas are still somewhat premature and perhaps not fully conclusive. I agree that it might not be appropriate to include them in the present paper. However, then I also think that statements in the text such as "the magnetic moment emanating from the spin-up channel Fe²⁺ → Fe³⁺ charge-transfer excitation" (lines 192/193) or "the increasing magnetic moment in the LuFe₂O₄ layer emanates from rising Fe²⁺ and Fe³⁺ density of states in the spin-up channel" (lines 209/210) are not very instructive. Also, the statement in lines 213-215, that one can understand this relation by considering the charge-ordered state in greater detail, is not very useful if this relationship is not further discussed in the present paper.*

Our reply: Referee 3 is correct that we write “We can understand in part why the enhanced magnetic moment emanates from the spin-up channel excitations by considering the charge-ordered state in more detail. But we are a little confused by the rest of the Referee’s comment. The very next line says “We do this below, but first...”, and we have an *entire section* on “Determining the charge ordering pattern in $(\text{LuFeO}_3)_3/(\text{LuFe}_2\text{O}_4)_1$ ”.

We are surprised by this confusion and suspect it arose due to the intervening discussion of interface design rules.

To clarify the situation, we moved the materials chemistry-related discussion of interface design rules to the Supplement. This greatly strengthens the connection between the charge ordering and the rest of the manuscript. We hope that Referee 3 will agree.

The other statements mentioned above are wholly experimental [Fig. 3]. They do not depend upon a particular model or detailed understanding of the charge-ordered state.

Referee 3 writes: *Regarding the discussion around Fig. 4, how the Lu distortion affects the orbital overlap and magnetic coupling, I still think that the discussion is rather superficial and not overly instructive. That the distortion changes the orbital overlap and that this will affect the magnetic coupling appears to be nearly trivial. However, why it actually seems to *enhance* the magnetic coupling (and thus the magnetic ordering temperature) remains unclear.*

Our reply: As mentioned in the prior question, we moved the materials chemistry-related discussion of interface design rules to the Supplement in order to provide greater continuity between the enhancement of the magnetic moment and our discussion of the charge ordering pattern. We continue to believe this section is useful for understanding the detailed pattern of Lu displacements + the impact of local lattice distortions on the interface with the iron double-layer but agree with the Referee that it is perhaps not essential in the main text.

Referee 3 writes: *The discussion of the different charge ordering pattern is still somewhat confusing. First, charge order patterns termed CO-I and CO-II are mentioned, then CO-FE and CO-DOPED are discussed. After having a look at the supplementary, it seems to me that CO-I and CO-II correspond to the LuFe_2O_4 parent compound, while the other two correspond to the superlattices (?). This should be clearly stated.*

Our reply: Yes, Referee 3 is 100% correct. CO-I and CO-II correspond to the parent LuFe_2O_4 compound, and CO-FE and CO-DOPED correspond to the superlattices. CO-FE and CO-DOPED are slight variations (subsets) of CO-II. We updated the text to clarify the hierarchy.

Referee 3 writes: *I appreciate the clarifications of the magnetic state used in the calculations. However one thing is still unclear to me. What does “antiferromagnetic alignment of the Fe^{3+} ” mean? Are they anti-parallel with respect to the Fe^{2+} moments or antiparallel with respect to each other (i.e, Fe^{2+} and half of the Fe^{3+} are “up”, and the rest of the Fe^{3+} “down”)?*

Our reply: Yes, good idea. We added the following text to the Methods section: “Specifically, in the Fe^{3+} - Fe^{3+} - Fe^{2+} layer of the LuFe_2O_4 block, the Fe^{3+} centers are anti-parallel to each other; one of the Fe^{3+} spins is up, and the other is down. The Fe^{2+} ’s are in different layers and align ferromagnetically.”

Referee 3 writes: *In summary, I think that with a few more clarifications the manuscript is in principle suitable for publication. I also think it provides clear progress in understanding this specific class of superlattices, which are indeed important examples of “engineered multiferroics”. As mentioned, I don’t see any clear strategies for improving the properties of other multiferroic systems following from this work. I assume that the method of separating the different contributions to the MCD can also be useful for other systems, but whether it’s a “breakthrough method” I don’t know. It certainly depends on the specific electronic structure whether it is possible to have a clear energy separation between the different contributions. I am therefore not sure whether the manuscript is specifically suited for Nature Communications, but I hope that the editor will be able to decide this based on the given information.*

Our reply: We thank Referee 3 again for his/her actionable feedback and comments about suitability of our work for publication in *Nature Communications*. We certainly agree that the $(\text{LuFeO}_3)_m(\text{LuFeO}_4)_n$ superlattices are an important class of engineered multiferroics and that we have made substantial progress in understanding the microscopic origin of high temperature magnetism and the specific charge ordering pattern - both of which we trace to interface effects. This does not mean that there is not more to learn, but our work demonstrates beyond a doubt how it is crucial for spectroscopists and materials growth teams to pivot toward the measurement of microscopic properties that theorists can not only model but use to create materials descriptors. Interfaces like domain walls and interface materials like the superlattices in this work have been interesting our team for a long time. The challenge has always been how, exactly, can we extract what is happening there. This led us to the use of near field infrared nano-spectroscopies for exploring domain wall dynamics and to the spectroscopic decomposition of magnetic circular dichroism presented here. In both cases, the findings dovetail with theory in exciting new ways, allowing us to probe and test the properties of interface materials like never before. Referee 3 is correct in noting that separating different contributions of the magnetic circular dichroism is easier (or harder) depending on the target material. As noted in our manuscript, the $(\text{LuFeO}_3)_m(\text{LuFeO}_4)_n$ superlattices certainly do not fall into the category of having a cleanly separated electronic structure. Instead, they sport six different types of Fe-related excitations, with many of them overlapping. In retrospect, we should have selected a simpler platform with which to test these ideas before applying them to $(\text{LuFeO}_3)_m(\text{LuFeO}_4)_n$. On the other hand, “necessity is the mother of invention”, and that is exactly what happened here. Referring to our spectroscopic decomposition method as a “break-through” technique is inappropriate and certainly my fault. I apologize and agree that the terminology should be avoided for any number of reasons. That said, we are immensely excited about the prospects that the spectroscopic separation + decomposition method provides. We hope that Referee 3 agrees.

Reviewer #3 (Remarks to the Author):

In the second revised version of the manuscript, the authors have clarified the remaining issues raised in my second report. The role of the Lu displacements and the different charge-order patterns are now clearly introduced. The other rearrangements have, in my opinion, also improved the clarity and focus of the manuscript. I still don't fully understand the meaning of a "magnetic moment emanating from the spin-up channel", but it seems to be clear from the context that the statement mainly refers to the corresponding changes in the MCD spectra. I think the paper is certainly suitable for publication. Regarding specific criteria for publication in Nature Communication, this was already addressed in my previous report. The editor should be able to easily judge this aspect.

Reviewer 3

Referee 3 writes: *In the second revised version of the manuscript, the authors have clarified the remaining issues raised in my second report. The role of the Lu displacements and the different charge-order patterns are now clearly introduced. The other rearrangements have, in my opinion, also improved the clarity and focus of the manuscript. I still don't fully understand the meaning of a "magnetic moment emanating from the spin-up channel", but it seems to be clear from the context that the statement mainly refers to the corresponding changes in the MCD spectra. I think the paper is certainly suitable for publication. Regarding specific criteria for publication in Nature Communication, this was already addressed in my previous report. The editor should be able to easily judge this aspect.*

Our reply: To address this issue, we modified the sentence in lines 209-212, page 8. The text now reads: "This behaviour demonstrates that increasing magnetic moment in the LuFe_2O_4 layer emanates from rising Fe^{2+} and Fe^{3+} density of states in the spin-up channel of the higher order superlattices. This conclusion emanates from the corresponding changes in the dichroic spectra." We hope that this modification clarifies the point.